# Reconfigurable spin current transmission and magnon–magnon coupling in hybrid ferrimagnetic insulators

Yan Li [1], Zhitao Zhang[2], Chen Liu[1], Dongxing Zheng [1], Bin Fang[1], Chenhui Zhang[1], Aitian Chen [1], Yinchang Ma [1], Chunmei Wang[2], Haoliang Liu [2] ✉, Ka Shen [3] ✉, Aurélien Manchon [4], John Q. Xiao[5], Ziqiang Qiu [6], Can-Ming Hu[7] & Xixiang Zhang [1] ✉

Coherent spin waves possess immense potential in wave-based information computation, storage, and transmission with high fidelity and ultra-low energy consumption. However, despite their seminal importance for magnonic devices, there is a paucity of both structural prototypes and theoretical frameworks that regulate the spin current transmission and magnon hybridization mediated by coherent spin waves. Here, we demonstrate reconfigurable coherent spin current transmission, as well as magnon–magnon coupling, in a hybrid ferrimagnetic heterostructure comprising epitaxial $Gd_3Fe_5O_{12}$ and $Y_3Fe_5O_{12}$ insulators. By adjusting the compensated moment in $Gd_3Fe_5O_{12}$, magnon–magnon coupling was achieved and engineered with pronounced anticrossings between two Kittel modes, accompanied by divergent dissipative coupling approaching the magnetic compensation temperature of $Gd_3Fe_5O_{12}$ ($T_{M,GdIG}$), which were modeled by coherent spin pumping. Remarkably, we further identified, both experimentally and theoretically, a drastic variation in the coherent spin wave-mediated spin current across $T_{M,GdIG}$, which manifested as a strong dependence on the relative alignment of magnetic moments. Our findings provide significant fundamental insight into the reconfiguration of coherent spin waves and offer a new route towards constructing artificial magnonic architectures.

Magnonics, which utilizes spin/magnon currents without electron flows, heralds an emerging paradigm for low-power information processing[1–3]. Spin waves, whose quasi-particles are magnons, serve as vital conduits for the transmission of spin/magnon currents. Various stimuli, like microwaves and thermal gradients, can excite coherent and incoherent spin waves, respectively[4,5]. Just like controlling charge currents in electronic devices, manipulating spin wave-mediated spin/magnon currents in magnonic devices is essential and yet challenging to enable their applications[6,7]. Recent progress has been made in operating incoherent spin waves, such as the spin colossal-magnetoresistance and magnon valve effect in insulator-based heterostructures[8–10]. Notably, coherent spin waves

[1]Physical Science and Engineering Division, King Abdullah University of Science and Technology (KAUST), Thuwal 23955–6900, Saudi Arabia. [2]Guangdong Provincial Key Laboratory of Semiconductor, Optoelectronic Materials and Intelligent Photonic Systems, School of Science, Harbin Institute of Technology (Shenzhen), 518055 Shenzhen, China. [3]The Center for Advanced Quantum Studies and Department of Physics, Beijing Normal University, 100875 Beijing, China. [4]Aix-Marseille Univ, CNRS, CINaM, Marseille, France. [5]Department of Physics and Astronomy, University of Delaware, Newark, Newark, DE 19716, USA. [6]Department of Physics, University of California at Berkeley, Berkeley, CA 94720, USA. [7]Department of Physics and Astronomy, University of Manitoba, Winnipeg, MB R3T 2N2, Canada. ✉e-mail: liuhaoliang@hit.edu.cn; kashen@bnu.edu.cn; xixiang.zhang@kaust.edu.sa

can encode information with their high fidelity through polarization, amplitude, and phase[11–15]. Moreover, their coherence also allows the establishment of a wave-based interference scheme, which potentially enables quantum computing via building up magnon–magnon coupling[16–20]. To facilitate coherent magnonic applications, methods to control the spin current transmission and magnon hybridization mediated by coherent spin waves must be developed[21–26]. However, there is currently a paucity of theoretical frameworks and structural prototypes, especially all-insulating architectures free from conduction bands, in this regard.

Ferrimagnetic and antiferromagnetic insulators offer strategic advantages in the development of programmable coherent magnonics owing to their exceptionally low magnetic dissipation, long spin/ magnon transmission length, absence of charge current, and the presence of multiple spin-wave polarizations[27–30]. Alternatively, compensated ferrimagnets composed of rare-earth and transition metal elements possess inequivalent magnetic sublattices with antiferromagnetic exchange interaction, thereby combining the characteristics of both ferromagnets and antiferromagnets[31]. Unlike antiferromagnets hosting resonant frequencies up to THz, the spin dynamics of compensated ferrimagnets are accessible through the microwave capabilities in the GHz regime[32,33].

In this work, we construct epitaxial hybrid ferrimagnetic heterostructures comprising the compensated ferrimagnet $Gd_3Fe_5O_{12}$ (GdIG) and the well-established ferrimagnetic insulator $Y_3Fe_5O_{12}$ (YIG). Such an architecture corresponds to a scenario where the non-magnetic element Y in the YIG layers is space-selectively substituted by the magnetic rare-earth element Gd. The Gd magnetic moment is sensitive to temperature, which provides a controllable degree of freedom to manipulate the magnetic configuration of the GdIG/YIG heterostructures. We identified a tunable magnon–magnon coupling as well as a divergent dissipative coupling under the variation of temperature, which were modeled by coherent spin pumping. Furthermore, we demonstrated, both theoretically and experimentally, coherent spin waves-mediated spin current with reconfigurable amplitudes in the YIG layer when the temperature passes through the magnetic compensation temperature of GdIG.

## Results and discussion

GdIG (74 nm)/YIG (46 nm) bilayers were deposited on (111)-oriented sGGG substrates using pulsed laser deposition, where the thickness of each layer is specified in parentheses [see Methods for details]. Additionally, two reference samples, GdIG (74 nm) and YIG (46 nm), were grown on (111)-oriented sGGG substrates. The crystal quality of the substrate/GdIG/YIG sample was characterized using scanning transmission electron microscopy (STEM). As shown in Fig. 1a and Supplementary Fig. 4b, atomic-resolution STEM images revealed a (111)-oriented epitaxial relationship among the sGGG substrate, GdIG layer, and YIG layer. This arrangement indicates the seamless continuity of the Fe sublattices across the GdIG/YIG interface, while the positions of the Gd element in the GdIG layers were replaced by Y atoms in the YIG layers.

Figure 1b displays the hysteresis loops of GdIG/YIG bilayers measured with an in-plane magnetic field $H$ at different temperatures,

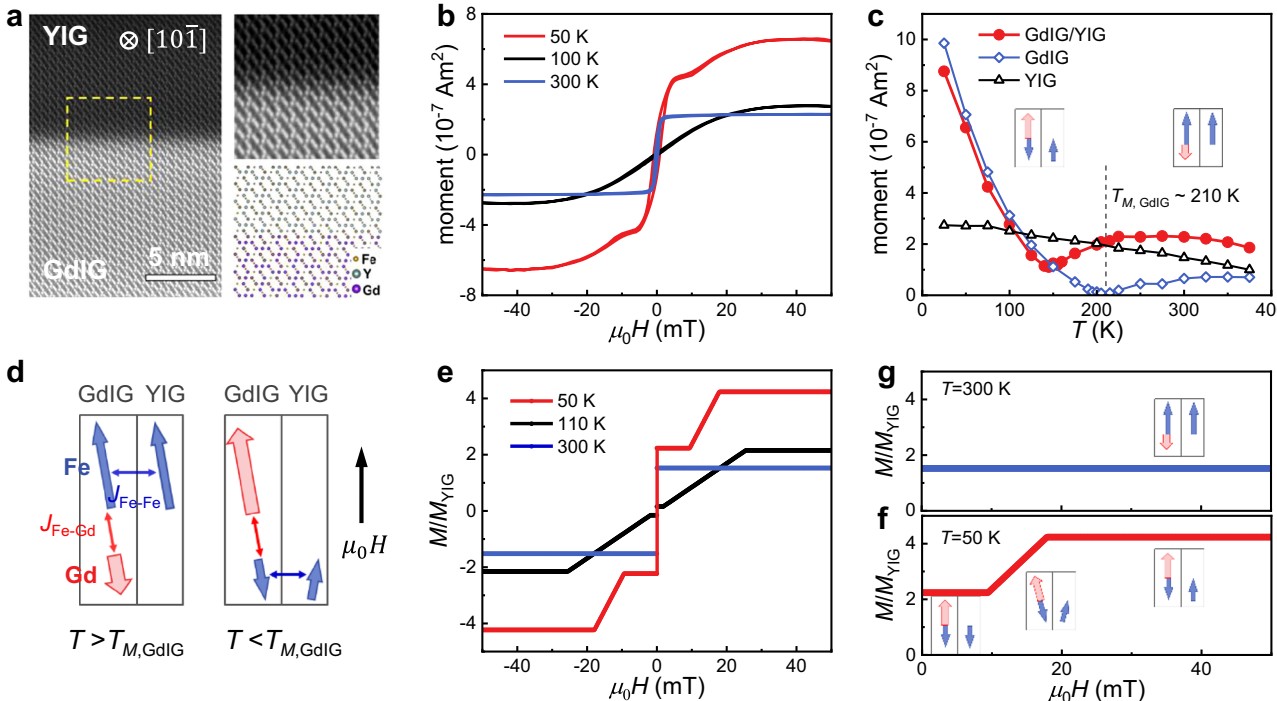

**Fig. 1 | Structural and magnetic properties of GdIG/YIG heterostructures.**
**a** Cross-sectional STEM image of the GdIG/YIG interface in the [10$\bar{1}$] plane. The top-right panel displays an atomic-resolution STEM image in zoomed-in view. The bottom-right panel showcases the corresponding atomic model of the GdIG/YIG interface with Fe (yellow), Gd (purple), and Y (green) atoms. **b** Magnetic hysteresis loops of GdIG/YIG plotted as a function of in-plane magnetic field at 50 K, 100 K, and 300 K, respectively. **c** Temperature dependence of magnetic moments extracted at $\mu_0 H = 50$ mT for the GdIG(74 nm), YIG(46 nm), and GdIG(74 nm)/ YIG(46 nm) samples. **d** Schematic illustration of the exchange coupling between

GdIG and YIG layers, with the interfacial Fe-Fe exchange interaction $J_{Fe-Fe}$ at the GdIG/YIG interface and the Gd-Fe exchange interaction $J_{Fe-Gd}$ in GdIG taken into account during the calculations. The right panel refers to the magnetic configuration at a large magnetic field that overcomes the interfacial coupling by aligning the net moments of GdIG and YIG in the same direction. The tetrahedral and octahedral Fe sublattices are treated as a single effective Fe magnetic sublattice. **e** Calculated hysteresis loops at various temperatures. **f, g** Enlarged portion of the calculated hysteresis loops at 300 K and 50 K, respectively. The insets depict the evolution of the magnetic configurations as a function of the applied in-plane magnetic field.

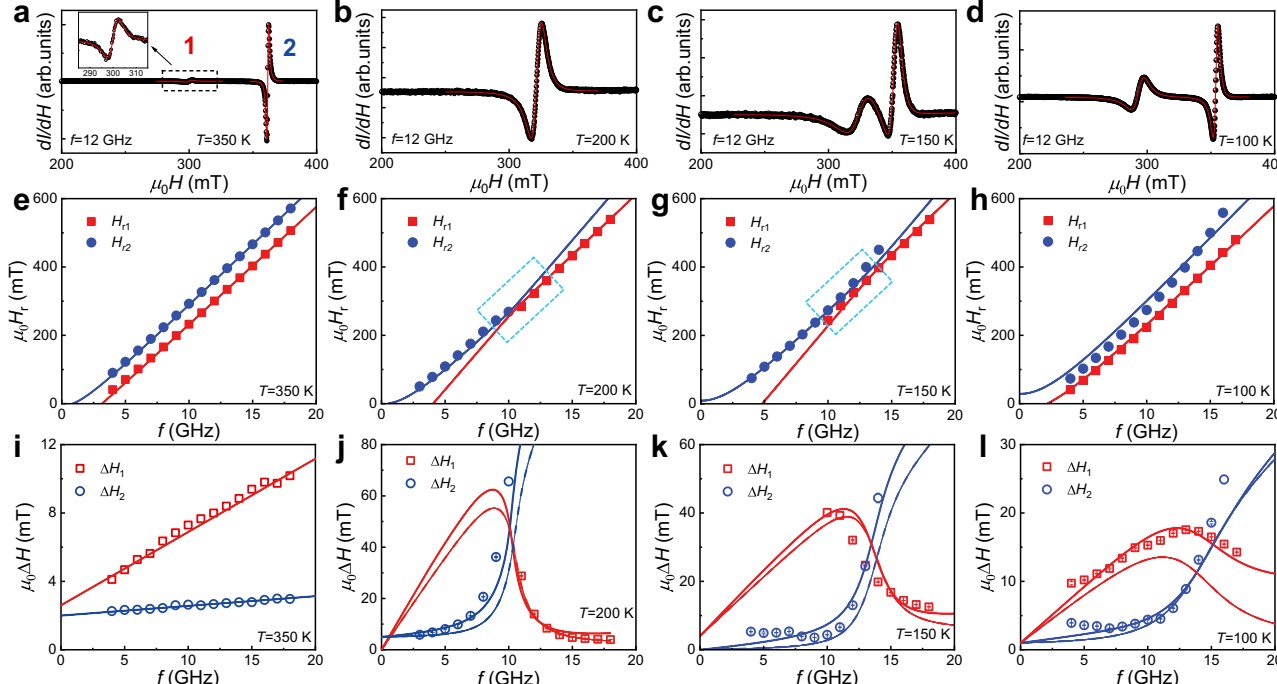

**Fig. 2 | Temperature evolution of magnon–magnon coupling and dissipative coupling. a–d** Derivative FMR absorption spectra measured at $f = 12\,\text{GHz}$. Two resonance modes are clearly identified at $T = 350\,\text{K}$, $150\,\text{K}$, and $100\,\text{K}$ but not at $200\,\text{K}$. These modes occurring at low and high magnetic fields are marked as mode 1 and mode 2, respectively. **e–h** Resonant field $H_r$ extracted from the FMR spectra plotted against the resonant frequency $f$ at $T = 350\,\text{K}$, $200\,\text{K}$, $150\,\text{K}$, and $100\,\text{K}$, respectively. Dashed rectangular boxes in **f** and **g** highlight the magnon–magnon coupling region. **i–l** Frequency dependence of the resonant linewidth $\triangle H$ extracted from the FMR spectra at $T = 350\,\text{K}$, $200\,\text{K}$, $150\,\text{K}$, and $100\,\text{K}$, respectively. The solid (dashed) lines show the fitting results with (without) coherent spin pumping. Error bars represent fitting uncertainty.

where the paramagnetic background of the substrate has been subtracted. The rectangular loop at 300 K exhibits a single switching step, indicating a characteristic of ferromagnetic exchange interaction between GdIG and YIG. The loop at 50 K exhibits a segmented switching feature, which is a typical feature of magnetization reversal in an antiferromagnetically coupled bilayer. Figure 1c presents the magnetic moment as a function of temperature for the GdIG, YIG, and GdIG/YIG samples, extracted from the hysteresis loops at $\mu_0 H = 50\,\text{mT}$. The net magnetic moment of the single GdIG layer vanishes at ~210 K, reflecting magnetic compensation of the Gd sublattice by the combined sublattices of strongly coupled tetrahedral and octahedral Fe sites. The net moment of GdIG is thus dominated by the Fe and Gd sublattices above and below the magnetic compensation temperature ($T_{M,\text{GdIG}}$), respectively[34,35]. In contrast, the magnetic moment of YIG shows a rather weak temperature dependence, owing to the non-magnetic nature of the Y element, which distinguishes it from the magnetic rare-earth element Gd present in GdIG. For the GdIG/YIG sample, the net moment reaches the minimum value at around 145 K, tens of Kelvin lower than $T_{M,\text{GdIG}}$ due to the interfacial exchange interaction.

In order to clarify the evolution of the magnetic configuration when temperature and applied magnetic field are varied, we performed numerical calculations based on a macrospin model. As schematized in Fig. 1d, we take into account the Gd-Fe antiferromagnetic interaction $J_{\text{Fe-Gd}}$ within GdIG, together with the effective interfacial Fe-Fe ferromagnetic exchange interaction $J_{\text{Fe-Fe}}$ between GdIG and YIG[36]. The lengths of the red and blue arrows in Fig. 1d are only relative and do not represent the specific value of the magnetic moments. Figure 1e displays the calculated hysteresis loops relative to the magnetization of YIG at several representative temperatures, which closely resemble the experimental results shown in Fig. 1b. Above $T_{M,\text{GdIG}}$, the net moment of

GdIG is dominated by the effective Fe lattices and is always aligned parallel to that of YIG throughout the entire applied magnetic field range, as illustrated in the inset of Fig. 1f. Below $T_{M,\text{GdIG}}$, the net moment of GdIG is dominated by the Gd sublattice, resulting in a field dependence of the magnetic configuration. At a low magnetic field, the effective Fe moment of YIG is parallel to that of GdIG, i.e., antiparallel to the moment of the Gd sublattice, as schematically drawn in the inset of Fig. 1g. As the applied magnetic field increases, a canting configuration is formed. Upon further increasing the magnetic field, both the Gd moment of GdIG and the Fe moment of YIG align along the applied magnetic field direction, whereas the moment of the effective Fe lattice in GdIG remains antiparallel to the Gd moment. That is, the interaction below $T_{M,\text{GdIG}}$ refers to the antiferromagnetic-like coupling between the YIG moment and the net moment of GdIG. More details on the variation of the magnetic configuration with the applied magnetic field are available in Supplementary Fig. 1 in Supplementary Information (SI). The good agreement between the calculated and experimental results suggests that the coupling between the GdIG and YIG layers can indeed be characterized by an effective ferromagnetic exchange interaction, predominantly mediated by the Fe-Fe exchange interaction (equivalent to an antiferromagnetic interaction between Gd in GdIG and Fe in YIG). The evolution of the magnetic configuration as a function of temperature makes GdIG/YIG a versatile platform for hybridizing magnons and reconfiguring spin current transmission.

To explore the coherent spin dynamics of GdIG/YIG films, we conducted in-plane ferromagnetic resonance (FMR) measurements[35,37]. Figure 2a–d displays the derivative spectra of FMR absorption at 12 GHz as a function of the magnetic field at 350 K, 200 K, 150 K, and 100 K, respectively. Except for the curve at 200 K, the spectra clearly show two resonant peaks with relatively small and large amplitudes, indicating the

presence of two resonant modes. These modes correspond to the exchange interaction-correlated Kittel modes dominated by the GdIG and YIG layers, respectively. We labeled the modes occurring at lower and higher magnetic fields as mode 1 and mode 2, respectively. The missing of the GdIG-dominated resonant peak at 200 K is attributed to its very weak intensity due to the near-zero magnetization and large damping of GdIG at temperatures close to $T_{M,GdIG}$. The resonant magnetic field $H_{r1}$ ($H_{r2}$) and linewidth $\Delta H_1$ ($\Delta H_2$) of mode 1 (mode 2) were extracted from the FMR spectra using the Lorentzian functions [see the details in SI]. Note that all FMR measurements were performed under magnetic fields exceeding 40 mT, where the net moments of both the GdIG and YIG layers saturate along the magnetic field direction, according to Fig. 1.

As shown in Fig. 2e–h, the variation of the magnetic configuration across $T_{M,GdIG}$ results in a change in the dispersion relation ($H_r$ versus $f$). At 350 K (above $T_{M,GdIG}$), the two observed resonances exhibit two well-separated branches, as shown in Fig. 2e. Similar results can be seen at temperatures above $T_{M,GdIG}$ [see Supplementary Fig. 5]. At 200 K around $T_{M,GdIG}$, it appears to exhibit solely an $H_r$-$f$ relation. However, there is a sudden shift in the $H_r$-$f$ curve around 10 GHz, as illustrated by the dashed rectangular box in Fig. 2f. At 150 K, a pronounced anticrossing feature between the two resonant modes is observed, as highlighted by the dashed rectangular box in Fig. 2g, which is a clear indication of strong magnon–magnon coupling[38]. Here, we did not extract $H_r$ and $\Delta H$ at some frequencies at 150 K and 200 K due to the extremely weak and almost indiscernible amplitudes of the peaks. These weak signals can be primarily attributed to the disparity in the magnetic damping parameter and magnetization between GdIG and YIG close to $T_{M,GdIG}$, as well as the mode hybridization, which further enables the transfer of energy between magnon excitations in two magnetic subsystems [see Supplementary Figs. 3 and 8]. Upon reducing the temperature to 100 K (below $T_{M,GdIG}$), there appeared to be slight level repulsion around 5 GHz in the $H_r$ versus $f$ curves of Fig. 2h.

Figure 2i–l plots the linewidth $\Delta H$ against the resonant frequency $f$ for the two modes at the corresponding temperatures. At 350 K (Fig. 2i), $\Delta H$ scales linearly with $f$ for both modes. Such a linear dependence can be observed at all temperatures above $T_{M,GdIG}$ [see Supplementary Fig. 5]. The different slopes of the two curves suggest the distinct damping parameters of these modes. Nevertheless, the relationship between $\Delta H$ and $f$ becomes nonlinear near and below $T_{M,GdIG}$, as shown in Fig. 2i–l. Specifically, at 200 K and 150 K, as $f$ increases, $\Delta H_1$ decreases while $\Delta H_2$ increases, particularly in the hybrid regime. At 100 K, $\Delta H_1$ increases with $f$ up to 13 GHz, and then decreases as $f$ further increases. Conversely, $\Delta H_2$ almost monotonically increases with frequency $f$ at 100 K.

To explain the above observation, the spin dynamics in GdIG/YIG are modeled using the coupled Landau–Lifshitz–Gilbert (LLG) equations[39,40]:

$$\frac{d\mathbf{M}_i}{dt} = -\mu_0\gamma_i\mathbf{M}_i\times(\mathbf{H}_i^{eff}+A_{ij}^{eff}\mathbf{M}_j)+\frac{\alpha_i}{M_i}\mathbf{M}_i\times\frac{d\mathbf{M}_i}{dt}+\frac{\Delta\alpha_i^{SP}}{M_i}\mathbf{M}_i\times\frac{d\mathbf{M}_i}{dt}$$
$$-\frac{\Delta\alpha_{ij}^{SP}}{M_j}\mathbf{M}_j\times\frac{d\mathbf{M}_j}{dt} \quad (1)$$

Here, $\gamma_i$, $\alpha_i$, $\mathbf{M}_i$, and $\mathbf{H}_i^{eff}$ are the gyromagnetic ratio, intrinsic Gilbert damping, magnetization, and effective magnetic field of the $i$-layer, respectively, and the subscript $i$ (or $j$) refers to the GdIG or YIG layers. $\Delta\alpha_{ij}^{SP}$ is damping due to coherent spin pumping into layer $i$ from layer $j$ [see SI]. In Eq. (1), two different mechanisms couple the dynamics of bilayers: (i) a field-like coupling (with a real coupling rate) caused by the effective exchange interaction $A_{ij}^{eff}=\frac{J_{eff}}{\mu_0 M_i M_j t_i}$, which is determined by the effective exchange coupling constant $J_{eff}$ and the thickness $t_i$ of the $i$-layer; (ii) a damping-like coupling

(with an imaginary coupling rate) caused by coherent spin pumping, which also exhibits the enhanced damping $\Delta\alpha_i^{SP}$. These two types of coupling can be intuitively understood from the classical example of two pendulums coupled by a spring force and a friction force[41]. Engineering the interplay of such coherent and dissipative couplings in magnonics has been of continuous and broad interest, as demonstrated in all-metallic ferromagnetic layers[40,42–44], all-metallic synthetic antiferromagnet[45], pure antiferromagnets[18], pure ferrimagnets[28], hybrid metal/insulator[46], metallic spin torque nano-oscillators[47], and cavity-magnonics[48]. However, until now, it has never been achieved using all-insulating heterostructures to engineer both couplings combined with magnon hybridization, where pure spin currents would be free from the parasitic effects of incoherent spin currents from conduction bands. Below, we demonstrate that both types of couplings are created in our all-insulating GdIG/YIG bilayers, which provide significant fundamental insight into the reconfiguration of coherent spin waves. The solution to the coupled LLG equations requires the determination of the eigenvalues of a matrix equation [see SI for details]. The real and imaginary parts of these eigenvalues are related to the resonance frequency and Gilbert damping, respectively[46,49,50].

This model fits well and satisfactorily reproduces the corresponding experimental $H_r$ versus $f$, as plotted by the solid lines in Fig. 2e–h. Particularly, the model effectively captures the magnon–magnon coupling, as evidenced by the pronounced anticrossing between the two Kittel modes at 150 K. Additionally, the $H_r$-$f$ curve with a sudden shift around 10 GHz at 200 K is also well reproduced with a subtle anticrossing feature. Furthermore, we have also conducted experiment to obtain the vector network analyzer transmission spectra. These spectra visually illustrate the characteristics of magnon–magnon coupling at 150 K and 200 K [see Supplementary Fig. 8]. The anticrossing gaps, that is the frequency splitting at the minimal resonance separation in the fits, are used to evaluate the magnon–magnon coupling strength, yielding values of 0.27 GHz (at $\mu_0 H = 269.3$ mT) at 200 K, and 0.50 GHz (at $\mu_0 H = 376.4$ mT) at 150 K. These values are comparable to the coupling strength of ~0.2 GHz between the Kittel mode and the perpendicular standing spin-wave mode in the metal-insulator YIG/Co hybrid system[17]. Previous works reported the behavior of magnon–magnon coupling between the Kittel modes in all-metallic synthetic antiferromagnets, such as CoFeB/Ru/CoFeB[45], as well as in antiferromagnets such as $CrCl_3$[18]. In such symmetric systems with identical magnetic layers, microwaves can excite acoustic and optical modes of opposite parities. These modes are constrained by the two-fold rotational symmetry, which results in a degeneracy at the crossing point, rather than magnon hybridization between the two modes. A tilting magnetic field is usually applied to lift the degeneracy. This further results in level repulsion of the dispersion relations with an anticrossing gap. In our all-insulating GdIG/YIG bilayers, the effective exchange interaction between the GdIG and YIG layers is on the order of ~1 mJ/m² and antiferromagnetic-like below $T_{M,GdIG}$. This was verified by the fitting of the effective exchange interaction, as illustrated in Fig. 3a, which coincides with the experimental and theoretical results in Fig. 1b, e. Such a temperature-sensitive antiferromagnetic-like interaction and the net moment of the Gd sublattices provide a great opportunity for engineering magnon hybridization. By adjusting the temperature, the $H_r$ versus $f$ curves of the two Kittel modes can intersect at 150 K and 200 K. This intersection is exemplified by the evolution of $H_{r1}$ and $H_{r2}$ at 10 GHz with temperature, as shown in Fig. 3b. Approaching $T_{M,GdIG}$ from above, $H_{r1}$ and $H_{r2}$ move apart. In contrast, approaching $T_{M,GdIG}$ from below, $H_{r1}$ and $H_{r2}$ move closer quickly. These distinct behaviors are attributed to the difference in the thickness, exchange interaction, and effective magnetization between the GdIG and YIG layers, which break the two-fold rotational symmetry and lead to the

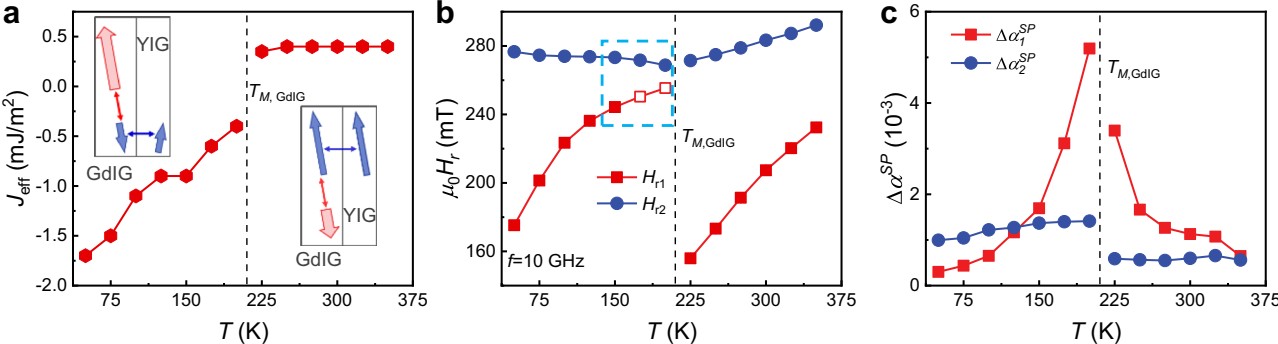

**Fig. 3 | Variation of magnetic parameters across the magnetic compensation temperature of GdIG. a** Temperature dependence of the effective exchange constant extracted from the fittings in Fig. 2. **b** Extracted resonant field $H_r$ of the two coupled modes as a function of temperature at $f = 10$ GHz. The solid and open symbols are obtained from experiments and fitting, respectively. **c** Temperature dependence of the enhanced damping induced by coherent spin pumping between the two hybrid modes.

hybridization of the two modes and the emergence of an anticrossing gap.

To explore the dissipative coupling between the resonant modes, we utilized the same parameters as those used for the fitting of the $H_r$ versus $f$ curves, to fit the frequency-dependent $\Delta H$. The solid and dashed lines in Fig. 2j–l correspond to the fitting outcomes with and without the coherent spin pumping term ($\frac{\Delta \alpha_i^{SP}}{M_i} \mathbf{M}_i \times \frac{d\mathbf{M}_i}{dt} - \frac{\Delta \alpha_{ij}^{SP}}{M_j} \mathbf{M}_j \times \frac{d\mathbf{M}_j}{dt}$), respectively. It is worth noting that while the fitting results without coherent spin pumping significantly deviate from the experimental observations, as exemplified by Fig. 2l, those incorporating coherent spin pumping do exhibit excellent agreement with our experimental observations both above and below $T_{M,GdIG}$. This unambiguously confirms the presence of coherent spin pumping in the coupled dynamics of the GdIG and YIG layers. Figure 3c summarizes the fitted values of $\Delta \alpha_1^{SP}$ and $\Delta \alpha_2^{SP}$ as a function of temperature. As the temperature approaches $T_{M,GdIG}$, there is a significant increase in $\Delta \alpha_1^{SP}$, which manifests the characteristic divergence of the dissipation with temperature and represents spin pumping out of GdIG. In contract, $\Delta \alpha_2^{SP}$, representing spin pumping out of YIG, exhibits only a weak dependence on temperature. The divergent dissipation in GdIG can be deduced from the contribution of spin pumping from the effective Fe and Gd lattices[51–53]. The effective Gilbert damping in the ferrimagnet GdIG is expressed by $\alpha_{eff0} \approx \frac{\alpha_{Fe} L_{Fe} + \alpha_{Gd} L_{Gd}}{|L_{Fe} - L_{Gd}|}$, where $L_{Fe}$ ($L_{Gd}$) and $\alpha_{Fe}$ ($\alpha_{Gd}$) represent the angular momentum and Gilbert damping of the Fe (Gd) sublattice, respectively[54]. Since the 4 f shell-dominated moments in Gd are more localized than the 3d shell-dominated moments in Fe, the effective Fe lattices play a dominant role in the coherent spin pumping. Assuming an enhancement $\Delta \alpha_{Fe,1}^{SP}$ in the damping of the effective Fe lattices due to coherent spin pumping, $\Delta \alpha_1^{SP}$ can be simplified as $\Delta \alpha_1^{SP} \approx \frac{\Delta \alpha_{Fe,1}^{SP}}{|L_{Fe} - L_{Gd}|}$[35]. Considering that the compensation temperature of the angular momentum ($L_{Fe} = L_{Gd}$) is approximately equal to $T_{M,GdIG}$ in GdIG[55], the above expression indeed gives the divergence of $\Delta \alpha_1^{SP}$ close to $T_{M,GdIG}$. The efficiency of the coherent spin pumping-induced dissipative coupling between the two Kittel modes can be parameterized by the spin mixing conductance $g_{mix}^{\uparrow\downarrow} = \frac{4\pi M_i t_i}{\gamma_i \hbar}$[45,46]. The enhanced damping yields $g_{mix}^{\uparrow\downarrow} = (1.7 - 4.8) \times 10^{19}$ m$^{-2}$, which is comparable to the values observed in classical ferromagnetic metal/heavy metal interfaces and the YIG/Py hybrid system[46,56]. Therefore, the coherence of spin pumping manifests itself as a divergent dissipative coupling with temperature in the ferrimagnetic all-insulating GdIG/YIG heterostructure, which characterizes a reconfigurable spin current

transmission mediated by the two coherent spin waves at the GdIG/YIG interface.

To further explore the spin transfer in the ferrimagnetic hybrid system, we performed spin pumping experiments to measure spin excitations into the YIG layer, as schematically drawn in Fig. 4a. Here, a 5 nm-thick Pt layer was deposited on GdIG/YIG to serve as a detector of spin current. Under the FMR condition for the GdIG/YIG/Pt sample, coherent magnetic precession is triggered. After spin excitations across the GdIG/YIG bilayer, the spin angular momentum carried by the coherent spin waves is converted into electric signals in the Pt layer due to the inverse spin Hall effect (ISHE)[57]. Figure 4b, d presents the FMR spectra as a function of the applied magnetic field at a fixed microwave frequency of 6 GHz and at two representative temperatures above and below $T_{M,GdIG}$, respectively. The corresponding ISHE voltage signals $V_{ISHE}(H)$ were recorded at a microwave power of 79.4 mW and exhibited a resonance Lorentzian shape with two peaks, as shown in Fig. 4c, e. Notably, the observed $V_{ISHE}(H)$ adheres to the resonance conditions, where the two extreme values of $V_{ISHE}(H)$ correspond to the resonant magnetic fields in Fig. 4b, d. Specifically, close to $T_{M,GdIG}$, it is difficult to simultaneously identify the two peaks of $V_{ISHE}(H)$ due to the large magnetic damping of the two modes [see Supplementary Figs. 3 and 8]. Furthermore, $V_{ISHE}(H)$ undergoes a sign reversal when sweeping positive (black) and negative (red) magnetic fields, thus further confirming the ISHE origin under FMR conditions. Moreover, we extracted the amplitudes $V_i^{ISHE}$ of $V_{ISHE}(H)$ for mode $i$, ($i = 1, 2$). Both $V_1^{ISHE}$ and $V_2^{ISHE}$ are proportional to the power of microwave irradiation, consistent with the expected spin pumping induced-ISHE mechanism [see Supplementary Figs. 9 and 10].

To evaluate the distinction of spin transfer between the two resonant modes across $T_{M,GdIG}$, a ratio $\xi$ is defined as $\xi = \frac{V_1^{ISHE}}{P_1'} / \frac{V_2^{ISHE}}{P_2'}$, where $V_i^{ISHE}$ is normalized with the relative absorbed microwave power $P_i'$ ($i = 1, 2$). Figure 4f presents a statistical summary of $\xi$ at 6 GHz as a function of temperature. Remarkably, $\xi$ remains relatively constant at -1.5 above $T_{M,GdIG}$, while it drops abruptly to -0.3 below $T_{M,GdIG}$. Additionally, the temperature-dependent $\xi$ at other frequencies confirms a similar sharp transition across $T_{M,GdIG}$ [see Supplementary Fig. 12c]. That is, the values of $\xi$ are greater (less) than 1 above (below) $T_{M,GdIG}$ at all measured frequencies.

To unravel the underlying mechanism of spin transfer across $T_{M,GdIG}$, we calculated the spin amplitudes of the two modes at the YIG layer by taking into account the fact that the ISHE signals in Pt are generated by the spin injection directly from the YIG layer[58,59]. The calculation details can be found in SI. As illustrated in the inset of Fig. 4g, the Kittel modes above $T_{M,GdIG}$, triggered under FMR

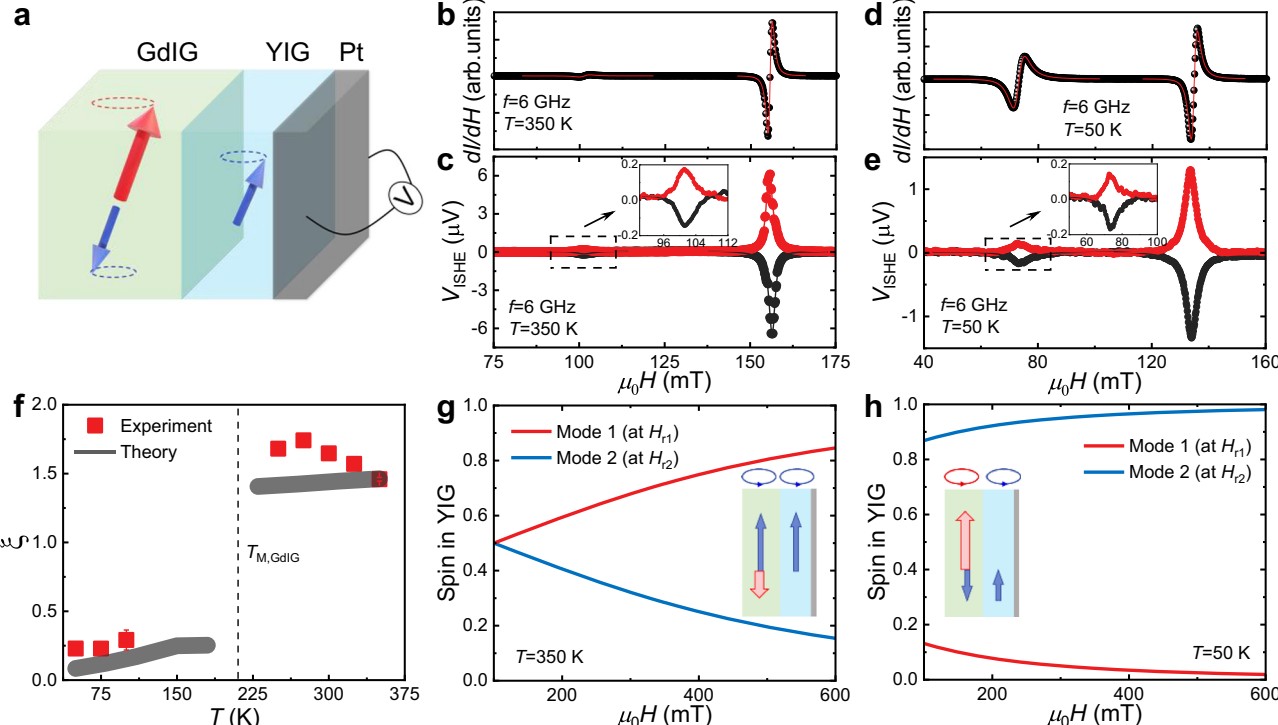

**Fig. 4 | Experiments and theory of reconfigurable spin current transmission.**
**a** Schematic of the spin pumping-driven ISHE measurements. Two resonant modes are excited by microwaves and the spin angular momentum carried by these modes is detected through ISHE voltages at the YIG/Pt interface. **b** Derivative FMR absorption at $f = 6$ GHz and $T = 350$ K. **c** Measured ISHE voltage $V_{ISHE}(H)$ as a function of the magnetic field at $f = 6$ GHz and $T = 350$ K. The black and red circles are the ISHE voltage signals acquired by sweeping positive and negative magnetic fields, respectively. The solid lines are fits, suggesting that the symmetric Lorentzian dominates the $V_{ISHE}(H)$ signals. The inset is a zoomed-in view of the hybrid mode 1. **d**, **e** Derivative FMR absorption (**d**) and the measured ISHE voltage $V_{ISHE}(H)$

(**e**) as functions of the magnetic field at $f = 6$ GHz and $T = 50$ K. **f** Experimental ratio of spin current transmission contributed by the two hybrid modes, along with the corresponding theoretical prediction, plotted as a function of temperature. Error bars represent fitting uncertainty. **g**, **h** Theoretical amplitudes of spin excitation contributed by the two coupled modes at the YIG/Pt interface as functions of the applied magnetic field at $T = 350$ K and 50 K, respectively. In the insets of **g**, **h**, the blue circles indicate the Fe sublattice-dominated precessions in the respective magnetic layers. Similarly, the red circle in the inset of **h** represents the Gd sublattice-dominated precession in its corresponding magnetic layer.

conditions in both layers, are Fe-dominated ones, which are strongly coupled by the interfacial ferromagnetic Fe-Fe exchange interaction. As a result of the parallel alignment of Fe moments, both modes 1 and 2 are highly hybrid modes of the two individual Kittel modes in the two layers. In particular, for the region of measurement in Fig. 4b–e (below ~200 mT), the dynamical spin amplitudes of the two resonant modes in the YIG layer become comparable with each other, as illustrated in Fig. 4g. With increasing applied magnetic field, the spin amplitude of YIG for mode 1 increases markedly, while that of mode 2 decreases because of the increasing frequency difference between the two Kittel modes in YIG and GdIG. This clearly elucidates the large value of $\xi$ above $T_{M,GdIG}$ in Fig. 4f. In contrast, the Kittel mode in GdIG is Gd-dominated below $T_{M,GdIG}$, and is only weakly coupled to the Fe-dominated mode in YIG due to the mismatch of the interfacially coupled Fe dynamics in the two layers, as sketched in the inset of Fig. 4h. As a result, modes 1 and 2 are primarily driven by the dynamics of Gd in GdIG and Fe in YIG, respectively, reflected by their difference in the spin amplitude in the YIG layer. As shown in Fig. 4h, the spin amplitude of mode 1 decreases with increasing magnetic field, while that of mode 2 increases. The limited dynamical spin component in YIG for mode 1 leads to reduced efficiency in spin injection into the Pt layer, which is equivalent to the suppression of spin transmission for mode 1 across the YIG layer. This effectively explains the significant reduction of $\xi$ when the temperature decreases across $T_{M,GdIG}$. To show the relevance of this mechanism quantitatively, we extracted the calculated ratio of the spin amplitudes of the two modes at the YIG/Pt interface at a resonant frequency of 6 GHz, and plotted the results in Fig. 4f, which

are in excellent agreement with our experimental observations. Our experimental and theoretical analysis suggest that the transmission of coherent spin waves strongly depends on the relative alignment of magnetic moments, resulting in a reconfigurable behavior in spin current transmission at the YIG/Pt interface.

In summary, a reconfigurable coherent spin-wave-mediated spin current transmission, as well as magnon–magnon coupling, has been realized in the compensated ferrimagnetic GdIG/YIG heterostructures. A magnon–magnon coupling between the two Kittel modes has been achieved close to $T_{M,GdIG}$, which potentially enables quantum engineering applications based on ferrimagnetic materials. This is accompanied by a divergent dissipative coupling induced by coherent spin pumping as the temperature approaches $T_{M, GdIG}$. Moreover, coherent spin waves propagating into the YIG layer exhibited reconfigurable amplitudes across $T_{M,GdIG}$, enabling control of the coherent spin waves-mediated spin current by virtue of the temperature-sensitive moments of rare-earth elements. These experimental results are in excellent agreement with the theoretical models. Our work opens new opportunities for operating coherent spin waves in ferrimagnetic magnonic architectures from both fundamental and applied viewpoints.

## Methods
### Sample preparation
GdIG, YIG, and GdIG/YIG samples were grown on (111)-oriented $Gd_{2.6}Ca_{0.4}Ga_{4.1}Mg_{0.25}Zr_{0.65}O_{12}$ (sGGG) substrates via pulsed laser deposition at a laser repetition frequency of 5 Hz. The substrate

temperature was set to 640 K, and the oxygen pressure was 85 mTorr during deposition. Subsequently, the samples were cooled to room temperature at a rate of 5 K/min. Afterward, a 5 nm-thick Pt layer was deposited at room temperature using a magnetron sputtering system. Magnetic property measurements were carried out using a Magnetic Property Measurement System (MPMS3, Quantum Design). The structures were characterized via STEM (Titan 80-300, FEI) after the sample was prepared by a focused ion beam (Helios 450, FEI).

### FMR and ISHE measurements

The experiments were conducted utilizing coplanar waveguide (CPW)-based setups in a Physical Property Measurement System (PPMS, Quantum Design). A Microwave Signal Generator (R&S®SMB100A) was used for microwave excitation. By sweeping the in-plane magnetic field, which was applied perpendicular to the signal line of the CPW, the FMR transmission signals modulated by a Helmholtz coil were picked up using Stanford Research SR830 lock-in amplifiers based on a phase-sensitive technique. The spin pumping-driven inverse spin Hall signals were measured using a Keithley 2182 meter.

### Theoretical calculation

The magnetic configuration and spin current transmission were calculated using a macrospin model. The magnon dispersion relation and dissipative coupling were modeled on the basis of the coupled LLG equations with coherent spin pumping. More details on the theoretical calculation are available in SI.

## Data availability

The data that support the findings of this work are presented in the main text and Supplementary Information. The source data used in the main text are available from the Figshare repository under https://doi.org/10.6084/m9.figshare.25188116.

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

## Acknowledgements

The work reported was funded by King Abdullah University of Science and Technology (KAUST), Office of Sponsored Research (OSR) under the Award Nos. ORA-CRG8–2019-4081 and ORA-CRG10-2021-4665. H.L.L. was supported by the National Natural Science Foundation of China (Grant No. 62174044) and the Regional Innovation Development Joint Program of the National Natural Science Foundation of China (Grant No. U22A20115). K.S. was supported by the National Natural Science Foundation of China (Grants No. 11974047 and 12374100) and the Fundamental Research Funds for the Central Universities.

## Author contributions

Y.L. conceived and designed this study. X.X.Z. supervised the project. Y.L. grew the insulating films. B.F. and A.T.C. deposited the Pt layers. C.L., D.X.Z., and C.H.Z. performed the cross-sectional TEM and analysis. Y.L. performed the magnetic moment, FMR, and ISHE measurements. Y.L., Z.T.Z., and C.M.W. analyzed the FMR data with assistance from H.L.L. K.S. performed the theoretical calculations. Y.L. wrote the manuscript, which was improved with very valuable input and comments from X.X.Z., H.L.L., K.S., A,M, J.Q.X., Z.Q.Q., C.M.H., and Y.C.M. All authors contributed to the discussion of the results and improvement of the manuscript.

## Competing interests

The authors declare no competing interests.
