## [Peer Review File · Nature Communications]

Reviewers' Comments:

Reviewer #2:

Remarks to the Author:

In the manuscript entitled "Reconfigurable coherent magnon transmission and magnon-magnon coupling in hybrid ferrimagnetic insulators", the authors experimentally and theoretically studied the FMR and spin pumping of GdIG/YIG heterostructures. An effective interfacial exchange coupling is identified. Because of the temperature-dependent net magnetic moment of GdIG, the spin pumping of the heterostructure can be tuned. The manuscript is well written and the research is intact and convincing. However, since Nature Communications aims to publish papers with "important advances of significance", I think the results presented in this manuscript is not significant enough. It may be better to be published in more specific journals, such as Communications Physics, Physical Review B, etc.

Particularly, I have several comments on the manuscript. (1) throughout the title, abstract and introduction of the manuscript, the authors emphasize the investigation and manipulation of magnons. But indeed only the FMR modes are studied. Of course, the FMR mode is a magnon with infinite wavelength in quantum mechanical point of view, but presenting the whole study using the word "magnon" is kind of exaggeration. (2) In "Reconfigurable coherent magnon transmission" section, the authors studied the spin pumping of the GdIG/YIG heterostructure. However, in FMR condition there is no "magnon current"---it is just a coherent precession and through spin pumping ISHE signal can be detected. The narration in this part is a little improper. (3) Although the classical LLG description is more suitable here, it is still necessary to quantize the spin wave to use the "magnon" language.

Reviewer #3:

Remarks to the Author:

The authors report on the measurement of spin dynamics in GdIG/YIG thin film heterostructures. The magnetization configuration of the compensated ferrimagnet GdIG is tuned by temperature and the magnetization dynamics of the heterostructure are measured by broadband ferromagnetic resonance (FMR) spectroscopy. The authors find that the FMR spectra depend strongly on temperature, which they interpret in terms of coupled macrospin dynamics.

The field of coupled spin dynamics in heterostructures is timely and relevant for a broad community, so the topic of this manuscript is suitable for Nature Communications. However, I am not convinced that the presented data fully supports the two main claims of the authors: 1 – magnon-magnon coupling, 2 – reconfigurable coherent magnon transmission. There are my specific concerns:

1 magnon-magnon coupling:

- In Line 141 the authors state that the two separate branches indicate strong exchange interaction. However, one would also get two FMR branches for two non-interacting layers with different anisotropy. So this specific data does not provide sufficient evidence for strong exchange interaction.
- The authors highlight in line 143 "a pronounced anticrossing feature" in Fig 2f. This is not in agreement with the resonance frequencies of the two modes visible in Fig. 2f and the corresponding raw data in Fig. 2 b. In Fig. 2b one observes only a single resonance, in agreement with the single data point at this frequency in Fig. 2f. For strongly coupled systems, there are always (for any frequency) two distinct resonance lines observed with corresponding anticrossing of the two modes in f vs H plots. The claim of strong magnon-magnon coupling is thus not supported by the data, at least not for this temperature.
- In Fig. 2c there are two distinct lines, but a quantitative evaluation of the coupling regime is missing – is the line separation minimized for 12 GHz? Does the line with stronger intensity move from higher field to lower field when changing frequency from, e.g., 10 GHz to 15 GHz (above and below the alleged hybridization point)? How can the authors otherwise know that they observe avoided crossing and not two uncoupled resonance lines with similar dispersions?
- To evaluate the coupling strengths and coupling regime, the authors need to compare the

coupling strength (minimal separation of resonance frequencies of the two modes in Fig. 2f – here apparently very close to 0) to the frequency linewidths of both resonances. Strong coupling is only achieved if the coupling strength is much larger than both linewidths.

- The authors show linewidths in Fig. 2i-l, but the interpretation here is very difficult because they show field linewidths at fixed frequency. When keeping frequency fixed but changing external magnetic field, the magnetic configuration is changing with field strength. Thus, the field linewidths cannot be directly interpreted in terms of magnon lifetime but are probably impacted by static reorientation of the magnetization(s). One would need to evaluate the frequency linewidths at fixed external magnetic field instead to discuss magnon-magnon coupling strengths.

2 reconfigurable magnon-magnon transmission:

- The model used by the authors assumes a macrospin ($k=0$) for each layer. I thus cannot see how it can be used to demonstrate magnon ($k \neq 0$) transmission? Transmission implies propagation of the excitation with some velocity, but indeed, one would expect the group velocity of the uniform mode measured here to be zero. So how can a uniform mode with zero velocity be transmitted? Why cannot the data in Fig. 4 be interpreted by resonant excitation of YIG magnetization (large peak in V_{ISHE}) vs forced excitation of YIG magnetization at the GdIG resonance by interfacial coupling (small peak in V_{ISHE})? One would expect that the strength of the interfacial coupling changes at T_{M} , GdIG as depicted in Fig. 3a.

- If the authors want to hold up the claim of reconfigurable magnon-magnon transmission, they should clearly discuss what the differences are between magnon-magnon transmission and coupling of two macrospins.

Other point:

- In the conclusions the authors state that the magnon-magnon coupling close to T_{M} , GdIG can potentially enable quantum entanglement applications. Please explain how this should work.

The manuscript may be suitable for Nature Communications if the authors can clarify these points in a satisfactory manner and revise the manuscript correspondingly.

We would like to thank the reviewers for their insightful and constructive comments/suggestions that have helped us improve our manuscript. In the following, we have addressed all comments thoroughly.

Reviewer #2 (Remarks to the Author):

In the manuscript entitled “Reconfigurable coherent magnon transmission and magnon–magnon coupling in hybrid ferrimagnetic insulators”, the authors experimentally and theoretically studied the FMR and spin pumping of GdIG/YIG heterostructures. An effective interfacial exchange coupling is identified. Because of the temperature-dependent net magnetic moment of GdIG, the spin pumping of the heterostructure can be tuned. The manuscript is well written and the research is intact and convincing. However, since Nature Communications aims to publish papers with “important advances of significance”, I think the results presented in this manuscript is not significant enough. It may be better to be published in more specific journals, such as Communications Physics, Physical Review B, etc.

Response: We sincerely thank you for reading our manuscript and for your very valuable comments. We are pleased that you appreciate that “the manuscript is well written and the research is intact and convincing”. Here, we would like to further highlight the specific advances achieved in our work.

Coherent spin waves/magnons are garnering increased attention due to their high fidelity and ultra-low energy consumption in device applications, as well as foundational importance in spintronics, magnonics, and quantum information science. This surge in interest is evident by the very recent publications¹⁻⁷. However, theoretical frameworks and structural prototypes, particularly those involving all-insulating architectures free from conduction bands, for controlling coherent spin waves are scarce. In our work, 1) Magnon-magnon coupling, accompanied by divergent dissipative coupling, was achieved and engineered with pronounced anticrossings between two Kittel modes by leveraging the temperature-sensitive magnetic moment of the Gd sublattices in all-insulating ferrimagnetic architectures. Such a possibility has not been demonstrated before. 2) A remarkable variation of amplitude in the coherent spin wave-mediated spin currents across the magnetic compensation point, relevant to the relative alignment of magnetic moments, was firstly demonstrated, both theoretically and experimentally. It establishes a novel approach for reconfiguring coherent spin waves using compensated ferrimagnetic insulators. Our findings, encompassing points 1) and 2), pave the way for wave-based devices that bridge spintronics with quantum

information science by constructing artificial magnonic architectures. In the following, we will delve into a comprehensive elucidation of the novelty and significance of these achievements.

[Redacted]

1) Reconfigurable magnon–magnon coupling in hybrid ferrimagnetic insulators.

Hybrid quantum magnonics, which combines spintronics and quantum information science, has emerged as a rapidly increasing field on quantum engineering/computing^{2,8,9}. In such systems, information is carried by magnons and processed through their coherent coupling with other quasiparticles such as photons, phonons, and magnons themselves. Synthetic antiferromagnets and layered antiferromagnets are served as a newly emerging host for magnon–magnon coupling owing to their high spin densities and potentially strong coupling^{1,10-12}. Taking synthetic antiferromagnets as an example, the coupled resonant modes are typically referred to as acoustic and optical modes under the FMR condition. The resonant frequencies of these modes exhibit different magnetic field dependence, allowing them to intersect in a certain field range [see the dash lines in Fig. R1b] when these modes are constrained by the two-fold rotational symmetry. That is, these modes degenerate at the crossing point. The broken two-fold rotational symmetry enables the interaction between the two modes, subsequently lifting this degeneracy. This results in level repulsion of the dispersion relations with an anticrossing [see the solid lines in Fig. R1b], which is well-known in the community of hybrid quantum magnonics as magnon-magnon coupling.

Engineering coherent coupling is an important prerequisite for expanding their functionality in hybrid quantum magnonic systems. In our work, we demonstrated the

tunability of magnon-magnon hybridization in insulating ferrimagnetic heterostructures. Magnon-magnon coupling is disabled/enabled by the temperature-sensitive moments in the compensated ferrimagnet GdIG, and the coupling strength ($2g$) reaches up to 0.50 GHz at 150 K. Furthermore, the tunability of magnon-magnon hybridization is accompanied by divergent dissipative coupling driven by coherent spin pumping approaching the magnetic compensation temperature of GdIG. **So far, it has never been achieved using all-insulating ferrimagnetic architecture** to engineer both coherent and dissipative couplings combined with magnon hybridization. **Our findings demonstrate that the all-insulating magnonic architectures involving compensated ferrimagnets open new avenues for optimizing magnon-magnon coupling and provide invaluable insights into the dissipation in magnon hybridization.**

2) Coherent spin waves-mediated spin current in hybrid ferrimagnetic insulators.

Current prototypes of spintronic/magnonic devices largely rely on incoherent spin waves to control spin currents or magnon currents, thereby achieving information transmission and processing. Such examples include the spin colossal-magnetoresistance and magnon valve effect based on thermal magnons^{13,14}. However, there's a growing shift in the spintronics community, moving from incoherent towards coherent spin waves or magnons. This trend is evident in recent studies, such as coherent and antiferromagnetic spin pumping¹⁵⁻¹⁷, spin transmission through phase-correlated magnons^{18,19}, and coherent magnon-induced domain-wall motion³. In comparison to incoherent spin waves, coherent spin waves can encode information with higher fidelity by their polarization, amplitude, and phase. This opens doors to establishing wave-based interference circuits and the potential for quantum computing.

In our work, **we discovered the crucial role of coherent spin pumping in modeling the frequency-dependent linewidth in the GdIG/YIG system without a Pt layer**, as shown in Figs. 3(j-l) in the main text. Furthermore, to explore the transfer of spin angular momentum (or spin current) between the two layers across all-insulating ferrimagnetic architecture, 5 nm-thick Pt layer was deposited on GdIG/YIG (forming GdIG/YIG/Pt) to serve as a detector of spin. When subjected to FMR conditions, **there was a pronounced variation in the amplitude of spin currents mediated by the two coherent spin waves across the magnetic compensation point. This change is significantly influenced by the relative alignment of the magnetic moments.** All of our observations were also theoretically modeled. We believe that **we have pioneered a novel approach to manipulate coherent spin waves by utilizing two distinct ferrimagnetic insulators.**

To meet the criteria, we have further emphasized the novelty and significance of our work and modified the part of abstract and introduction in the revised manuscript. Our paper focused on magnon-magnon coupling in the original manuscript, not just spin pumping, even though we utilized conventional FMR measurements. For a clearer depiction of magnon-magnon coupling, we've enriched the revised supplementary information (SI) with VNA-FMR spectra and theoretical analysis. Moreover, according to your comments, we realize that the use of the term “magnon transmission” in the original manuscript might be misleading although we previously used “transmission” in order to emphasize the transfer of spin angular momentum between the two layers. In the revised manuscript, we use “spin current mediated by the coherent spin wave” and “spin current transmission” instead of the term “magnon transmission”. We do hope that our response, additional experiments, and calculations will inspire a more favorable outlook on the significance of our work. We believe therefore that this will affirm the suitability of our paper for publication in Nature Communications. Below, we address your questions and comments on a point-by-point basis.

1. Particularly, I have several comments on the manuscript. (1) throughout the title, abstract and introduction of the manuscript, the authors emphasize the investigation and manipulation of magnons. But indeed only the FMR modes are studied. Of course, the FMR mode is a magnon with infinite wavelength in quantum mechanical point of view, but presenting the whole study using the word “magnon” is kind of exaggeration.

Response: Thank you for this feedback. In the original manuscript, when we discussed magnon-magnon coupling, we employed the term “magnon”. As shown by the solid and dash lines in Fig. R1b, the broken two-fold rotational symmetry can lift the degeneracy of the resonant modes at the crossing point, which results in level repulsion of the dispersion relations with an avoided crossing (or anticrossing) [see the solid lines in Fig. R1b]. In the community of hybrid quantum magnonics, this phenomenon is referred to as magnon-magnon coupling. In keeping with this traditional term, we believe that the reviewer will not object to our use of the word “magnon” when discussing “magnon-magnon coupling”.

Moreover, we primarily used the term “magnon transmission” to describe the transfer of spin angular momentum (or spin current) between the two layers. To avoid any potential misunderstanding about the zero velocity of the FMR modes, we use “spin current mediated by the coherent spin wave” and “spin current transmission” in place of using “magnon transmission” in the revised manuscript. We also welcome any specific suggestions to further refine this terminology and enhance the clarity of the manuscript.

2. In “Reconfigurable coherent magnon transmission” section, the authors studied the spin pumping of the GdIG/YIG heterostructure. However, in FMR condition there is no “magnon current”---it is just a coherent precession and through spin pumping ISHE signal can be detected. The narration in this part is a little improper.

Response: Thanks for your comment and feedback. In the “Reconfigurable magnon–magnon coupling and coherent spin pumping” section, we found that it is essential to consider the contribution of coherent spin pumping to accurately model the frequency-dependent linewidth in the **GdIG/YIG system without a Pt layer**, as shown in Figs. 3(j-l) in the main text. The coherent spin pumping from the coupled magnetization dynamics drives the transfer of spin angular momentum carried by spin current, as shown in Fig. R2, resulting in an additional damping. That is, the coherent spin pumping driven dissipative coupling (or damping) with a divergence reflects the loss of spin angular momentum carried by spin current in the all-ferrimagnetic-insulator heterostructure GdIG/YIG bilayer, without the need for a Pt layer.

Fig. R2 Schematic diagram of coherent spin pumping accompanied with spin current.

To further explore the transfer of spin angular momentum mediated by the two coherent spin wave modes between the two layers across the GdIG/YIG structure, a 5 nm-thick Pt layer was deposited on GdIG/YIG, **forming the GdIG/YIG/Pt system**, which we discuss in the “Reconfigurable coherent spin waves-mediated spin current transmission” section. If the precession in GdIG/YIG pumps spin currents into the Pt layer (used as a spin current detector), the ISHE voltages, corresponding to different modes, will be induced in the Pt layer. These ISHE signals reflect the spin amplitudes of the two coherent spin wave modes in the YIG layer, considering that ISHE signals in Pt result from direct spin injection from the YIG layer. This means that the spin pumping-driven ISHE voltage experiment mirrors the two coherent spin waves-mediated spin current within the GdIG/YIG system.

In response to your comment, we’ve opted to use “spin current” in place of the primary term “magnon current” in the revised manuscript.

3. Although the classical LLG description is more suitable here, it is still necessary to quantize the spin wave to use the “magnon” language.

Response: Thanks for your suggestion. In the community of hybrid quantum magnonics, current understanding of the magnon-magnon coupling physics largely relies on the macrospin approximation. Indeed, there is a need for the development of a quantum perspective. In the revised manuscript, for the sake of consistency with established terminology in the previous work, we retain “magnon-magnon coupling” to describe the level repulsion of the dispersion relations f - H with anticrossings. Furthermore, we used “spin current mediated by coherent spin waves” instead of “magnon transmission”. Such adjustments aim to unify all descriptions within the macrospin model framework in the main text.

Additionally, for the convenience of readers interested in the quantum language descriptions, we also use a quantum framework to describe spin excitation and magnon-magnon coupling in the revised SI (lines 45-56 and lines 73-76). We employed an effective Hamiltonian to elucidate spin excitation,

$$H_m = A_1 a_{Fe'}^\dagger a_{Fe'} + A_2 a_{Fe}^\dagger a_{Fe} + A_3 a_{Gd}^\dagger a_{Gd} + (D_1 a_{Fe}^\dagger a_{Fe'} + D_2 a_{Fe}^\dagger a_{Fe'}^\dagger + D_3 a_{Fe}^\dagger a_{Gd}^\dagger + h.c.).$$

Here, $a_{Fe'}$, a_{Fe} , and a_{Gd} are annihilation operators of spin excitation for S_{Fe}^{YIG} , S_{Fe}^{GdIG} , and S_{Gd}^{GdIG} , respectively. We also rewrite the microscopic spin model in terms of the creation (\hat{a}^\dagger and \hat{b}^\dagger) and annihilation (\hat{a} and \hat{b}) operators to describe magnon-magnon coupling:

$$H = \hbar\omega_{0a} \left(\hat{a}^\dagger \hat{a} + \frac{1}{2} \right) + \hbar\omega_{0b} \left(\hat{b}^\dagger \hat{b} + \frac{1}{2} \right) + \hbar(g\hat{a}\hat{b}^\dagger + g^*\hat{a}^\dagger\hat{b}),$$

where ω_{0a} and ω_{0b} are the uncoupled magnon modes, and g is the coupling strength.

Reviewer #3 (Remarks to the Author):

The authors report on the measurement of spin dynamics in GdIG/YIG thin film heterostructures. The magnetization configuration of the compensated ferrimagnet GdIG is tuned by temperature and the magnetization dynamics of the heterostructure are measured by broadband ferromagnetic resonance (FMR) spectroscopy. The authors find that the FMR spectra depend strongly on temperature, which they interpret in terms of coupled macrospin dynamics.

The field of coupled spin dynamics in heterostructures is timely and relevant for a broad community, so the topic of this manuscript is suitable for Nature Communications. However, I am not convinced that the presented data fully supports the two main claims of the authors: 1 – magnon-magnon coupling, 2 – reconfigurable coherent magnon transmission. There are my specific concerns:

Response: Thank you for your insightful comments and acknowledgment of the timeliness and relevance of our topic. In the following, we will address your concerns one by one.

1. magnon-magnon coupling:

1.1 • In Line 141 the authors state that the two separate branches indicate strong exchange interaction. However, one would also get two FMR branches for two non-interacting layers with different anisotropy. So this specific data does not provide sufficient evidence for strong exchange interaction.

Response: Many thanks for pointing out the issue. We fully agree with you. Indeed, the two well-separated branches at 350 K is not sufficient to indicate the strong exchange interaction between the GdIG and YIG layers. The interaction should be supported by additional data. Actually, the M - H loops derived from both experiments and calculations, as shown in Figs. 1b and 1e, validate a strong exchange interaction between GdIG and YIG. Additionally, when comparing the f - H_r of the single GdIG film, it becomes necessary to introduce an exchange interaction term (effective exchange interaction constant up to 0.4 mJ/m² at 350 K) for GdIG/YIG to simultaneously fit the f - H_r curves of both mode 1 and mode 2, as depicted in Fig. R3. These data should support our claim. However, to soften the argument, we have removed the incorrect statement “indicating the strong exchange interaction between the GdIG and YIG layers” from the description to 350 K in the revised manuscript.

Fig. R3 Resonant field H_r extracted plotted against the resonant frequency f at $T=350$ K for the GdIG and GdIG/YIG samples.

1.2 • The authors highlight in line 143 “a pronounced anticrossing feature” in Fig 2f. This is not in agreement with the resonance frequencies of the two modes visible in Fig. 2f and the corresponding raw data in Fig. 2 b. In Fig. 2b one observes only a single resonance, in agreement with the single data point at this frequency in Fig. 2f. For strongly coupled systems, there are always (for any frequency) two distinct resonance lines observed with corresponding anticrossing of the two modes in f vs H plots. The claim of strong magnon-magnon coupling is thus not supported by the data, at least not for this temperature.

Response: We apologize for the misleading and abrupt statement in the previous version. Instead of the two resonant peaks observed at 350 K and 100 K, **only a distinct resonant line was evident at 200 K. This is attributed to the insufficient intensity of the other peak even in high-resolution experiments.** Generally, the intensity of FMR signals is approximately proportional to $\frac{tM_{net}}{\Delta H}$, where t , ΔH , and M_{net} denote the thickness, linewidth and net magnetization of magnetic layer, respectively. As shown in Fig. R4a (or Fig. 1c in the main text), the net magnetization M_{net} of the single GdIG film almost vanishes at the $T_{M,GdIG}$ of ~ 210 K. As the temperature approaches $T_{M,GdIG}$, the net magnetization of GdIG decreases, while the damping (or linewidth) increases^{11,20,21}. Consequently, it is very difficult to capture the resonant peak at 200 K even for the single GdIG film of nanoscale thickness, as shown in Fig. R4d. In the case of the GdIG/YIG bilayer, for the sake of simplification, the two resonant lines can be interpreted as the exchange interaction-correlated Kittel modes dominated by the GdIG and YIG layers, respectively, as shown in Fig. R4b. As a result, it is a significant challenge to experimentally capture the exchange interaction-correlated Kittel modes dominated by GdIG (see the red block in Fig. R4b) at 200 K for the GdIG/YIG bilayers due to the very weak intensity of the corresponding FMR peak.

Referring to Fig. R4c, which provides a detailed view of Fig. 2f from the main text, there is a clearly sudden shift in f - H around 10 GHz at 200 K. This hints at the possibility of two f - H lines. To further clarify this point, we also conducted FMR measurements based on vector network analyzer (VNA-FMR). As shown in Figs. R4e and f, the VNA-FMR spectrum at 200 K clearly indicate a “discontinuity” in f - H and a relatively weak FMR intensity around 10 GHz, which is a characteristic (i.e., anticrossing feature) of magnon-magnon coupling. That is, the manifestation of magnon-magnon coupling leads to a redistribution of precession energy, reflected in the FMR spectrum. The anticrossing feature is more obvious in the VNA-FMR spectrum at 150 K, as shown in Fig. R8. Such an anticrossing feature is ubiquitous and often observed when magnons hybridize with other quasiparticles, such as photons, phonons, and magnons themselves, as shown in Fig. R5.

Fig. R4 **a**, Temperature dependence of magnetic moments for the GdIG, YIG and GdIG/YIG samples, as seen in Fig. 1c of the main text. **b**, Schematic of magnon-magnon coupling. The dashed lines represent the Kittel modes dominated by GdIG and YIG without magnon-magnon coupling. The solid lines depict hybrid modes, exhibiting magnon-magnon coupling at the anticrossing point. The intensity of the corresponding peaks marked by the red block is weak due to the small magnetization and large damping of GdIG at 200 K (near $T_{M, \text{GdIG}}$ of $\sim 210 \text{ K}$). **c**, Enlarged Fig. 2f from the main text, depicting the f - H relations at 200 K. **d**, Field-derivative of the vector network analyzer transmission spectra $\partial S_{21}/\partial H$ (VNA-FMR spectrum) as a function of magnetic field and frequency at 200 K for the single GdIG film. No distinct resonance line (f - H line) was observed due to the very small magnetization and large damping of GdIG close to $T_{M, \text{GdIG}}$. **e**, VNA-FMR spectrum for GdIG/YIG at 200 K. **f**, Enlargement of Fig. R4e.

[Redacted]

To further shed more light on the characteristics of the magnon-magnon coupling, we calculated the FMR spectrum at 200 K. As depicted in Fig. R6a, the FMR intensity varies with frequency, showcasing the redistribution of precession energy in hybrid systems. We further identified the resonant peaks calculated at 6 GHz, highlighted in Fig. R6b. Significantly, the intensity of peak 2 surpasses that of peak 1 by over 80 times. This elucidates the difficulty in clearly detecting peak 1 at 200 K in our experiments. Consequently, it is not always possible to clearly observe two distinct resonance lines at all frequency points for our system with the compensated ferrimagnet GdIG. Drawing from the data in Fig. R6a, we illustrated the calculated VNA-FMR spectrum, as shown in Fig. R6c, which well reproduces the sudden shift in f - H around anticrossing point shown in Fig. 4c, and closely aligns with the experimentally obtained VNA-FMR spectrum (see Fig. R4e). As a result, these experiments and calculations all support the existence of magnon-magnon coupling at 200 K.

In response to this comment, we have updated the description of the resonant peaks corresponding to Fig. 2f in the revised manuscript (lines 143-154). The revised text acknowledges the existence of magnon-magnon coupling at 200 K. However, we have opted not to assert that this coupling at 200 K is strong, given the difficulty of simultaneously detecting two peaks in the experiment (lines 189-197). Furthermore, we have also added the VNA-FMR results and theoretical calculations in the revised SI (lines 170-186, and lines 116-131).

Fig. R6 a, Calculated FMR peaks at various frequencies at 200 K for GdIG/YIG. **b**, Calculated FMR peak at $f=6$ GHz and 200 K. **c**, FMR spectrum based on the calculated FMR data presented in Fig. R6a. The solid lines represent the peak positions.

1.3 • In Fig. 2c there are two distinct lines, but a quantitative evaluation of the coupling regime is missing – is the line separation minimized for 12 GHz? Does the line with stronger intensity move from higher field to lower field when changing frequency from, e.g., 10 GHz to 15 GHz (above and below the alleged hybridization point)? How can the authors otherwise know that they observe avoided crossing and not two uncoupled resonance lines with similar dispersions?

Response: Thank you for your valuable comments. Based on both experimental data and fitted results presented in Fig. 2c, we assessed the coupling region. As illustrated in Fig. R7a, from a frequency standpoint, there is a minimized separation at ~ 13.4 GHz between the two f - H line. Fig. R7b also illustrates a minimum frequency separation (anticrossing gap of ~ 0.5 GHz) between the f - H lines. Additionally, the nonlinear variation of linewidth with frequency, as depicted in Fig. 2k, also verifies two coupled resonant lines.

Fig. R7 a, Separation of the two f - H lines (Fig. 2c in the main text) from a frequency standpoint at 150 K. **b**, Separation of the two f - H lines from a magnetic field perspective at 150 K.

Moreover, we have also conducted the VNA-FMR measurements at 150 K to assess the changes in the intensity of the f - H lines. As shown in Fig. R10, the intensity of mode 2 decreases as the resonant frequency increases, whereas the intensity of mode 1 increases

with the rising resonant frequency. This observation with the redistribution of precession energy indicates the presence of coupling between these two resonant modes. This behavior is notably distinct from the uncoupled scenario, where the intensity of the peak should gradually decrease with an increase in resonant frequency.

To address your insightful comments, we have added the VNA-FMR spectra in the updated SI (lines 162-187), which enhance the visual representation of magnon-magnon coupling. Additionally, we have enriched the SI with both a discussion and the calculated FMR spectra related to magnon-magnon coupling (lines 117-132 in the SI).

Fig. R8 Measured VNA-FMR spectrum as a function of magnetic field and frequency for GdIG/YIG at 150 K.

1.4 • To evaluate the coupling strengths and coupling regime, the authors need to compare the coupling strength (minimal separation of resonance frequencies of the two modes in Fig. 2f – here apparently very close to 0) to the frequency linewidths of both resonances. Strong coupling is only achieved if the coupling strength is much larger than both linewidths.

Response: Thank you for this meaningful comment. As you pointed out, the magnon-magnon coupling strength is typically assessed by evaluating the minimal separation/gap ($2g$) in the frequency axis between the two modes around the anticrossing point. To determine whether the coupling is strong ($C > 1$) or weak ($C < 1$), the system cooperativity is defined as $C = \frac{g^2}{\kappa_1 \kappa_2}$, where κ_1 and κ_2 represent the frequency linewidths of mode 1 and 2, respectively. Thus, when the separation between two peaks exceeds the linewidths of both resonant peaks, it becomes easier to extract the coupling strength.

As we addressed in the response to your comment #1.2, it is extremely difficult to simultaneously distinguish the two resonant peaks at 200 K (near $T_{M,GdIG} \sim 210$ K). Therefore, it is very hard to determine the coupling strength at 200 K solely based on the experimental observations. In this work, we roughly determined the anticrossing

gap $2g \approx 0.27$ GHz by integrating the experiment and calculation at 200 K. Moreover, the field-modulated FMR and VNA-FMR measurements at 150 K indicate a strong magnon-magnon coupling with $2g \approx 0.50$ GHz and $C \approx 7$. In the revised manuscript, we just state the existence of a strong magnon-magnon coupling at 150 K. However, we have opted not to assert that magnon-magnon coupling at 200 K is strong.

1.5 • The authors show linewidths in Fig. 2i-l, but the interpretation here is very difficult because they show field linewidths at fixed frequency. When keeping frequency fixed but changing external magnetic field, the magnetic configuration is changing with field strength. Thus, the field linewidths cannot be directly interpreted in terms of magnon lifetime but are probably impacted by static reorientation of the magnetization(s). One would need to evaluate the frequency linewidths at fixed external magnetic field instead to discuss magnon-magnon coupling strengths.

Response: Thanks for your comment. To assess the magnetic configuration, we carried out the spin Seebeck effect (SSE) measurements, a bulk-sensitive magnetometry technique. This method enables us to effectively remove the influence of the paramagnetic background from the substrate when compared to the traditional M - H measurements. As shown in Fig. R9b, the measured SSE signals (V_{SSE}) saturate when the applied magnetic field surpasses ~ 50 mT. Moreover, we also calculated the changes in the magnetic configuration with varying magnetic fields, as illustrated in Fig. R10 [Figs. 1(e-f) of the main text and Figs. S1(a-c) in the SI]. As observed, when the applied magnetic field exceeds ~ 40 mT, the magnetic moment becomes saturated. Both our experimental and calculated results support the presence of a stable magnetic configuration within the magnetic field range of ~ 50 -600 mT, which corresponds to the range employed in our FMR measurements. Consequently, we can reasonably conclude that the influence of magnetic configuration on the field linewidth is negligible.

Fig. R9 a, Schematic of the SSE measurements for GdIG/YIG. **b**, Measured SSE signals (V_{SSE}) as a function of the applied magnetic field at various temperatures.

Fig. R10 Evolution of magnetic configurations as a function of the applied in-plane magnetic field at various temperatures. **a-c**, Calculated angles θ_G and θ_Y as functions of the applied magnetic field at 250 K, 150 K, and 50 K, respectively. The inset in **a** is the definition of θ_G and θ_Y .

Furthermore, sweeping the frequency under various magnetic fields allows us to obtain FMR spectra and simultaneously determine the frequency linewidths. However, due to the frequency and temperature dependence of microwave transmission, acquiring high-resolution frequency linewidths is technically challenging [referenced in Rev. Sci. Instrum. 89, 076101 (2018)]. Moreover, Rev. Sci. Instrum. 89, 076101 (2018) has demonstrated that the Gilbert damping obtained in the field domain is equivalent to that in the frequency domain. Therefore, our FMR experiments employed the field-modulated FMR technique to extract the field linewidth in the main text. Compared to the linewidth measurement based on the sweeping frequency, the accuracy of the field-modulated FMR measurements is commonly higher when simultaneously extracting the resonance field and resonance linewidth. This method, employed by the commercial company NanOsc, has also gained recognition in studies on coupled systems, such as in Phys. Rev. Lett. 124, 117202 (2020) and Sci. Adv. 5, eaax9144 (2019).

To address this comment, we added the SSE results into the revised SI (lines 238-252).

2. reconfigurable magnon-magnon transmission:

- *The model used by the authors assumes a macrospin ($k=0$) for each layer. I thus cannot see how it can be used to demonstrate magnon ($k \neq 0$) transmission? Transmission implies propagation of the excitation with some velocity, but indeed, one would expect the group velocity of the uniform mode measured here to be zero. So how can a uniform mode with zero velocity be transmitted?*

Why cannot the data in Fig. 4 be interpreted by resonant excitation of YIG magnetization (large peak in ViSHE) vs forced excitation of YIG magnetization at the GdIG resonance by interfacial coupling (small peak in ViSHE)? One would expect that the strength of the interfacial coupling changes at T_M , GdIG as depicted in Fig. 3a.

- *If the authors want to hold up the claim of reconfigurable magnon-magnon transmission, they should clearly discuss what the differences are between magnon-magnon transmission and coupling of two macrospins.*

Response: Thank you for your invaluable comments. In the original version, we intended to use the term “magnon transmission” to emphasize the transfer of spin angular momentum (or spin current) between the two layers. According to your comments and that of reviewer #2, we recognize that using the term “magnon transmission” might cause confusion. In the revised manuscript, we used “spin current mediated by the coherent spin waves” and “spin current transmission” instead of “magnon transmission”. Of course, we are always open to specific suggestions on how to refine the text to enhance the clarity of the manuscript.

Moreover, we actually used the coupling of two macrospins (as you pointed out) to illustrate the transfer of spin angular momentum between two layers in the original manuscript, rather than reconfigurable magnon-magnon transmission. We suspect that the insets in Figs. 4g-h may have caused some misunderstanding. In the insets of Figs. 4g-h, the blue circles indicate the Fe sublattices-dominated precessions in the respective magnetic layers. Similarly, the red circle in the inset of Fig. 4h represents the Gd sublattice-dominated precessions in its corresponding magnetic layer. We are sorry that this part is misleading and not clear in the original version. In our paper, we observed that the normalized ISHE signals dominated by GdIG are large although the interfacial coupling is small above $T_{M,GdIG}$. In contrast, these signals are subdued although the interfacial coupling is substantial below $T_{M,GdIG}$. The variations of the ISHE signals are described by the defined parameter ξ in the main text. Essentially, in the all-insulating heterostructure GdIG/YIG containing the compensated insulator GdIG, the variations in the magnetic configurations determine whether the Gd-Fe coupling or the Fe-Fe coupling is dominant in the dynamics, thus leading to the observed differences in the transfer of spin angular momentum between two layers above and below $T_{M,GdIG}$. Based on your comments, we have provided annotations for the insets in Figs. 4g-h (lines 532-534).

Other point:

- 3 • *In the conclusions the authors state that the magnon-magnon coupling close to $T_{M,GdIG}$ can potentially enable quantum entanglement applications. Please explain how this should work.*

Response: Thank you for the feedback. We first introduce the fundamentals of quantum computing. In quantum mechanics, the typical quantum state of a qubit is expressed as

a linear superposition of its two orthonormal basis states (basis vectors). These basis states are commonly denoted as $|0\rangle$ and $|1\rangle$. A pure qubit state is a coherent superposition of the basis states. This means that a single qubit can be described by a linear combination of $|0\rangle$ and $|1\rangle$,

$$|\psi\rangle = \alpha|0\rangle + \beta|1\rangle, \quad (\text{R1})$$

Here, α and β are the probability amplitudes and both are complex numbers. Essentially, a two-level system (qubit) serves as the foundational element in quantum engineering/computing, as shown in Fig. R11a. Essentially, many different systems can behave as driven qubits^{8,9,22,23}.

Next, we'll consider hybrid quantum magnonic system that possess magnon-magnon coupling between acoustic and optical modes. Taking this as an example, we will elucidate the application of magnon-magnon coupling in quantum computing. For such a hybrid system with magnon-magnon coupling, the complex eigenvectors for the hybrid mode can be expressed as^{10,24}:

$$\begin{pmatrix} A_o \\ A_a \end{pmatrix}_{\pm} = \frac{1}{\sqrt{g^2 + (\Delta_{\pm} \pm \sqrt{\Delta^2 + g^2})^2}} \begin{pmatrix} g \\ \Delta_{\pm} \pm \sqrt{\Delta^2 + g^2} \end{pmatrix}. \quad (\text{R2})$$

The wavefunction for the hybrid mode is a linear combination of $|o\rangle$ and $|a\rangle$, having complex amplitudes A_o and A_a , respectively. The eigenvectors detailed in Eq. (R2) are depicted in the representation ($|o\rangle$, $|a\rangle$). Therefore, the wavefunction can be rewritten as:

$$\emptyset = \begin{pmatrix} e^{i\delta} A_1 \\ A_2 \end{pmatrix} = A_o \begin{pmatrix} -1 \\ 1 \end{pmatrix} + A_a \begin{pmatrix} 1 \\ 1 \end{pmatrix}. \quad (\text{R3})$$

Eq. (R3) establishes a connection between the complex wave function hybridization and the observed parameters, including the amplitudes A_i ($i=1, 2$) and phase δ . In fact, as shown in Fig. R11b, Eq. (R3) can be also rewritten as the hybridization pseudo-spin state within a magnonic Bloch sphere,

$$|h\rangle = \cos \frac{\theta}{2} |o\rangle + e^{i\varphi} \sin \frac{\theta}{2} |a\rangle \quad (\text{R4})$$

Like Eq. (R1), Eq. (R3) and (R4) form the foundation of the hybrid magnonic system with magnon-magnon coupling for quantum computing.

Above, we have outlined the fundamental principles of the hybrid quantum magnonic system for quantum computing. However, there are still many technical challenges to overcome in the practical development of quantum computers based on the hybrid quantum magnonic system. In the original manuscript, we used the phrase ‘‘potentially enable quantum entanglement applications,’’ which might lead to some confusion. In the revised manuscript, we have changed it to ‘‘potentially enable quantum engineering

applications”. To better understand the two-level system GdIG/YIG, a 2x2 matrix has been added in the SI (lines 107-115). The coherent and dissipative couplings appear as the real and imaginary part of the complex non-diagonal matrix elements.

[Redacted]

The manuscript may be suitable for Nature Communications if the authors can clarify these points in a satisfactory manner and revise the manuscript correspondingly.

Response: We hope that the additional experiments and analyses, along with the updates in the revised manuscript, have already adequately address your concerns.

References

- 1 Comstock, A. H. *et al.* Hybrid magnonics in hybrid perovskite antiferromagnets. *Nat. Commun.* **14**, 1834 (2023).
- 2 Diederich, G. M. *et al.* Tunable interaction between excitons and hybridized magnons in a layered semiconductor. *Nat. Nanotechnol.* **18**, 23-28 (2023).
- 3 Fan, Y. *et al.* Coherent magnon-induced domain-wall motion in a magnetic insulator channel. *Nat. Nanotechnol.* **18**, 1000-1004 (2023).
- 4 Bae, Y. J. *et al.* Exciton-coupled coherent magnons in a 2D semiconductor. *Nature* **609**, 282-286 (2022).
- 5 Hortensius, J. R. *et al.* Coherent spin-wave transport in an antiferromagnet. *Nat. Phys.* **17**, 1001-1006 (2021).
- 6 Han, J., Cheng, R., Liu, L., Ohno, H. & Fukami, S. Coherent antiferromagnetic spintronics. *Nat. Mater.*, 1-12 (2023).
- 7 Makihara, T. *et al.* Ultrastrong magnon-magnon coupling dominated by antiresonant

- interactions. *Nat. Commun.* **12**, 3115 (2021).
- 8 Awschalom, D. D. *et al.* Quantum Engineering With Hybrid Magnonic Systems and Materials (Invited Paper). *IEEE Transactions on Quantum Engineering* **2**, 1-36 (2021).
- 9 Yuan, H. Y., Cao, Y., Kamra, A., Duine, R. A. & Yan, P. Quantum magnonics: When magnon spintronics meets quantum information science. *Physics Reports* **965**, 1-74 (2022).
- 10 MacNeill, D. *et al.* Gigahertz Frequency Antiferromagnetic Resonance and Strong Magnon-Magnon Coupling in the Layered Crystal CrCl₃. *Phys. Rev. Lett.* **123**, 047204 (2019).
- 11 Liensberger, L. *et al.* Exchange-Enhanced Ultrastrong Magnon-Magnon Coupling in a Compensated Ferrimagnet. *Phys. Rev. Lett.* **123**, 117204 (2019).
- 12 Shiota, Y., Taniguchi, T., Ishibashi, M., Moriyama, T. & Ono, T. Tunable Magnon-Magnon Coupling Mediated by Dynamic Dipolar Interaction in Synthetic Antiferromagnets. *Phys. Rev. Lett.* **125**, 017203 (2020).
- 13 Guo, C. Y. *et al.* A nonlocal spin Hall magnetoresistance in a platinum layer deposited on a magnon junction. *Nat. Electron.* **3**, 304-308 (2020).
- 14 Qiu, Z. *et al.* Spin colossal magnetoresistance in an antiferromagnetic insulator. *Nat. Mater.* **17**, 577-580 (2018).
- 15 Li, J. *et al.* Spin current from sub-terahertz-generated antiferromagnetic magnons. *Nature* **578**, 70-74 (2020).
- 16 Vaidya, P. *et al.* Subterahertz spin pumping from an insulating antiferromagnet. *Science* **368**, 160-165 (2020).
- 17 Li, Y. *et al.* Coherent Spin Pumping in a Strongly Coupled Magnon-Magnon Hybrid System. *Phys. Rev. Lett.* **124**, 117202 (2020).
- 18 Han, J. *et al.* Birefringence-like spin transport via linearly polarized antiferromagnetic magnons. *Nat. Nanotechnol.* **15**, 563-568 (2020).
- 19 Guckelhorn, J. *et al.* Observation of the Nonreciprocal Magnon Hanle Effect. *Phys. Rev. Lett.* **130**, 216703 (2023).
- 20 Li, Y. *et al.* Unconventional Spin Pumping and Magnetic Damping in an Insulating Compensated Ferrimagnet. *Adv Mater* **34**, e2200019 (2022).
- 21 Ng, I., Liu, R., Ren, Z., Kim, S. K. & Shao, Q. Survey of Temperature Dependence of the Damping Parameter in the Ferrimagnet Gd₃Fe₅O₁₂. *IEEE Transactions on Magnetics* **58**, 1-6 (2022).
- 22 Ivakhnenko, O. V., Shevchenko, S. N. & Nori, F. Nonadiabatic Landau–Zener–Stückelberg–Majorana transitions, dynamics, and interference. *Physics Reports* **995**, 1-89 (2023).
- 23 Zhang, X. A review of common materials for hybrid quantum magnonics. *Materials Today Electronics* **5** (2023).
- 24 Exploring wavefunction hybridization of magnon-magnon hybrid state. *arXiv:2308.14463*, 2023

Reviewers' Comments:

Reviewer #2:

Remarks to the Author:

In the revised manuscript, the authors modified some narrations to make them more proper. However, the essential question -- whether this is a current across the bilayer -- is not solved yet.

1. About the spin pumping between the layers. The authors used the behavior of FMR linewidth to claim the spin pumping between the layers. But the data in Fig. 2(j-l) is not significant enough. The deviation between the experimental data and model calculation cannot unambiguously make the statement that the solid line is better than the dashed line. Other mechanisms like three-magnon processes are not ruled out, either. To take a step back, I actually believe that the interlayer spin pumping exists. But Onsager reciprocal relation should apply here so that the current pumped from GdIG to YIG should be the same as the one back, otherwise physical explanation is needed.

2. About "spin current", no matter mediated by "magnon" or "spin wave". The FMR mode is excited in the whole sample, so this is "source" instead of a "current", whose intuitive definition is that some particles or excitations flow across an area with some velocity. If one can somehow excite FMR mode in one side (using a local field or spin Hall current, for example) and detect the spin current in the other side, then this is a spin current transmission. Even if this scenario is achieved, the tunable transmission across the heterostructure is not a surprise.

Some minor comments:

1. Page 7, line 180: "all" is duplicated?
2. Page 8, line 204: parity->parities?
3. Page 9, line 224: in the in-line equation, the second term should be α_j instead of α_i ?
4. Page 9, line 235-239, I do not quite understand this statement. I think it should be further explained.
5. Page 9, line 243, the citation needs brackets.

To summarize, I still think this manuscript is not suitable for Nature Communications.

Reviewer #3:

Remarks to the Author:

In response to the reports by both reviewers, the authors have revised the manuscript and SI. In the revised manuscript, they now refer to spin-current transmission instead of magnon transmission (it is now clear that they study FMR and not propagating magnons) and clarified previously confusing aspects of the data evaluation regarding the magnon-magnon coupling strength.

With the revisions made by the authors and their detailed reasoning provide in their reply, I believe that I have now been able to understand the physics at play. My previous concerns have thus been adequately addressed and resolved. I particularly appreciate the additions to the SI regarding determination of the coupling strength, which I now find well described and convincing. I now agree with the main conclusions about the observed physical phenomena drawn by the authors. I find that the revised manuscript is a valuable addition to the body of literature in the field. As all-insulating heterostructures are an important platform for spin dynamics, I have the impression that the topic of the work is suitable for Nature Comm. Most relevant for me, I have no further concerns regarding the interpretation, validity, and discussion of the results.

Reviewer #2 (Remarks to the Author):

In the revised manuscript, the authors modified some narrations to make them more proper. However, the essential question -- whether this is a current across the bilayer - is not solved yet.

Response: Thank you very much for your time and comments in improving our paper. In the following, we address all comments point-by-point. Changes and additions to the revised version and the supplemental information (SI) are marked yellow.

1. About the spin pumping between the layers. The authors used the behavior of FMR linewidth to claim the spin pumping between the layers. But the data in Fig. 2(j-l) is not significant enough. The deviation between the experimental data and model calculation cannot unambiguously make the statement that the solid line is better than the dashed line. Other mechanisms like three-magnon processes are not ruled out, either. To take a step back, I actually believe that the interlayer spin pumping exists. But Onsager reciprocal relation should apply here so that the current pumped from GdIG to YIG should be the same as the one back, otherwise physical explanation is needed.

Response: Thank you for your insightful comment. Based on your comment, we have assessed the possibility of three-magnon processes. If a parabolic magnon dispersion ($f = f_{FMR} + Ak^2$) is considered, the frequency f_{FMR} , corresponding to the FMR mode, is already the minimum value of frequency in the magnon dispersion, making it impossible to further split into two magnons. If we consider a more realistic magnon dispersion with magnetic dipole-dipole interactions, as shown in Fig. R1, three-magnon processes may occur. In this case, the FMR frequency is given by $f_{FMR} = \gamma\mu_0\sqrt{H(H + M)}$. The minimum magnon frequency is $f_{min} = \gamma\mu_0H$. The condition for the occurrence of three-magnon processes is $f_{FMR} > 2f_{min}$ ^{1,2}. Substituting the expressions above, this is equivalent to $H < M/3$. Taking the example at 100 K, the magnetizations of GdIG and YIG is approximately 240 mT. Therefore, the influence of three-magnon processes is possible only if $\mu_0H < 80$ mT. However, under the FMR condition at 100 K, as shown in Fig. 2d, the resonant magnetic fields (~50-600 mT) in our FMR measurements are mostly larger than 80 mT. Therefore, it can be inferred that three-magnon processes are not likely playing a significant role in our experiments. Moreover, in Fig. 2l, the deviation from experimental results becomes more pronounced for the curve without coherent spin pumping at higher frequencies (at higher magnetic fields), further emphasizing the insignificance of three-magnon processes. Additionally, in the context of two-magnon scattering, the FMR linewidth increases and saturates with increasing frequency^{3,4}. This is inconsistent with our experimental results where

the linewidth decreases with increasing frequency, as illustrated by the red curves in Fig. 2j-l.

In our work, we employed a model based on coherent spin pumping, which fits quite well and consistently reproduces the two experimental H_r versus f curves and two ΔH versus f curves at different temperatures. It is important to clarify that the distinction between solid and dashed lines may have been underestimated due to the large linewidth scale of the figures even up to 80 mT (Fig. 2j-k). However, as seen from the plot with a smaller scale of the y-axis (Fig. 2l), the red solid line does remarkably improve the fitting of the experimental results, where a clear difference between the red solid and dashed lines can be clearly recognized. For example, at 15 GHz, the difference between the two lines can reach up to ~40% of the experimental result. Therefore, we conclude that other mechanisms like three-magnon and two-magnon processes do not hold significance in our study, and our model based on coherent spin pumping is more convincing.

To further illustrate spin current or angular momentum transfer, let us consider a symmetric system NM1/FM1/NM/FM2/NM2 with identical magnetic layers for simplicity. In this case, as shown in Fig. R2, there are two collective modes under FMR conditions: a perfectly symmetric mode (or acoustic mode) in which the precessions of two magnetic layers are in-phase, and perfectly antisymmetric mode (or optical mode) in which the precessions are out-of-plane. **Despite the Onsager reciprocal relation ensuring equal efficiency for rightward and leftward spin currents in each magnetization precession, a phase difference exists between the precessions of two magnetic layers in the two collective modes.** As a result, the effective spin current from FM1 to FM2, induced by coherent spin pumping, is canceled due to the in-phase mutual precessions in the symmetric mode. However, the current is nonzero and amplified in the antisymmetric mode due to the out-of-phase mutual precessions of two FM layers. Such a spin current transmission associated with the phase of collective precession has been confirmed by our co-author Ziqiang Qiu's papers^{5,6} and other publications^{7,8}.

In our asymmetric system GdIG/YIG/Pt, collective modes and spin transmission are more complicated compared to the perfectly symmetric and antisymmetric modes in above symmetric system. We can comprehend the spin transfer between GdIG and YIG from a perturbative perspective. In fact, according to previous research, interlayer exchange coupling and spin pumping (also referred to as dynamic coupling) are considered compatible mechanisms, stemming from the time-retarded response of the interlayer exchange coupling⁸⁻¹⁴. The resonance of the GdIG magnetization under FMR

condition will force the precession in YIG due to the two couplings between them. Since the Pt layer is directly in contact with YIG in our GdIG/YIG/Pt stack, the detected ISHE signals at the YIG/Pt interface involve the current (or spin transmission) from GdIG to YIG. Conversely, if Pt were in contact with GdIG in a Pt/GdIG/YIG structure, the detected ISHE signals at the GdIG/Pt interface would involve the current from YIG to GdIG.

Following your comment, an assessment regarding three-magnon processes has been added into the revised SI [lines 163-175].

[Redacted]

Fig. R2 Symmetric mode with in-phase precessing magnetizations and antisymmetric mode with precessing magnetizations in a symmetric system NM1/FM1/NM/FM2/NM2. FM and NM are ferromagnet, non-ferromagnet, respectively. The exchange coupling between two FM layers can be tuned by thickness of the NM layer.

2. About “spin current”, no matter mediated by “magnon” or “spin wave”. The FMR mode is excited in the whole sample, so this is “source” instead of a “current”, whose intuitive definition is that some particles or excitations flow across an area with some velocity. If one can somehow excite FMR mode in one side (using a local field or spin Hall current, for example) and detect the spin current in the other side, then this is a spin current transmission. Even if this scenario is achieved, the tunable transmission across the heterostructure is not a surprise.

Response: Thanks for your comment. The spin current transmission is acknowledged by you in the following scenario: locally exciting FMR mode on one side and detecting the spin current in the other side. However, it is not acknowledged that spin current transmission takes place in the scenario where the FMR mode is excited in the whole sample. From a perturbative perspective, these two scenarios should be equivalent to some extent. Whether locally or whole excited, the resonance of the GdIG magnetization (resonant excitation) under FMR condition induces the precession of YIG (forced excitation) due to the interlayer exchange coupling and spin current-mediated dynamic coupling between them. In this case, the effective spin transmission from GdIG to YIG can be detected at the YIG/Pt interface. Conversely, the resonance of the YIG magnetization under FMR condition will also drag the precession of GdIG due to these couplings. Our experiments indicate that it is possible to achieve a spin current at the YIG/Pt interface by locally exciting the resonance of the GdIG side far from the GdIG/YIG interface (such excitation can be realized via the spin orbit torque-FMR). Similar approach, that the FMR mode is excited in the whole sample, have been widely adopted to investigate spin current and spin pumping in FM1/FM2 heterostructures and FM1/NM/FM2 systems with interlayer exchange coupling^{7,8,14,15}. In these examples, the pumped spin current at the interfaces can be expressed by spin excitations: $\mathbf{I}_s = \frac{\hbar}{e} \sum_{i,j} G_{ij} \mathbf{m}_i \times \dot{\mathbf{m}}_j$, where G_{ij} and \mathbf{m}_i denote the spin-mixing conductance matrix and the unit vector of the magnetic moment of sublattice i , respectively^{11,16-18}.

You may raise concerns about statements -- spin current **across the whole sample**. In response to these concerns, we further modified some narrations to make them more proper. For instance, “propagation of spin current across the GdIG/YIG bilayer” has been revised into “spin excitations across the GdIG/YIG bilayer” (line 257). In our manuscript, we assert that spin current driven by coherent spin pumping exists at the GdIG/YIG interface, and the transfer of spin angular momentum (spin current) at the YIG/Pt interface also takes place, manifesting as a strong dependence on the relative alignment of magnetic moments.

Moreover, despite some disputes in the use of the term “current”, it does not diminish the novelty of our work. Our research introduces a highly innovative approach for modulating spin transfer by leveraging compensated ferrimagnet. Furthermore, for the first time, magnon-magnon coupling between Kittel modes is constructed in an all-insulating YIG/GdIG heterostructure, that provides a new route for quantum computing based on magnon-magnon coupling.

Some minor comments:

1. Page 7, line 180: “all” is duplicated?

Response: Thank you for pointing out this issue. We have corrected this typo in the revised manuscript.

2. Page 8, line 204: parity->parities?

Response: Thank you. The word has been corrected in the revised manuscript.

3. Page 9, line 224: in the in-line equation, the second term should be α_j instead of α_i ?

Response: Thank you. For clarity, the second term is denoted as $\Delta\alpha_{ij}^{SP}$ in the revised manuscript, where $\Delta\alpha_{ij}^{SP}$ is damping due to spin pumping from layer j into layer i .

4. Page 9, line 235-239, I do not quite understand this statement. I think it should be further explained.

Response: Thanks for your comment. Unlike ferromagnets that have only one effective magnetic lattice, compensated ferrimagnets composed of rare earth and transition metal elements possess two magnetic sublattices at least. The effective Gilbert damping in the ferrimagnet GdIG is expressed as $\alpha_{eff0} \approx \frac{\alpha_{Fe}L_{Fe} + \alpha_{Gd}L_{Gd}}{|L_{Fe} - L_{Gd}|}$, where L_{Fe} (L_{Gd}) and α_{Fe} (α_{Gd}) represent the angular momentum and Gilbert damping of the Fe (Gd) sublattice, respectively¹⁹. Under coherent spin pumping, the total damping is described as $\alpha_{tot} = \alpha_{eff0} + \Delta\alpha_1^{SP} \approx \frac{(\alpha_{Fe} + \Delta\alpha_{Fe,1}^{SP})L_{Fe} + (\alpha_{Gd} + \Delta\alpha_{Gd,1}^{SP})L_{Gd}}{|L_{Fe} - L_{Gd}|}$, where $\Delta\alpha_1^{SP}$ is the enhanced damping induced by spin pumping in GdIG. In addition, $\Delta\alpha_{Fe,1}^{SP}$ and $\Delta\alpha_{Gd,1}^{SP}$ are the

enhanced damping of the effective Fe and Gd lattices due to coherent spin pumping. The magnetic moment of Gd mainly originates from the inner 4f shell-spin, which are more localized compared to the 3d shell-dominated moments in the Fe. The efficiency of spin pumping (or spin transfer) between 4f shell-spins and 3d shell-spins is much weaker than that of the Fe 3d shell-spins with itself or the conduction electrons. For example, the interfacial exchange coupling of the Gd 4f-spins with the Pt conduction electrons is weaker than that of the Fe 3d-spin²⁰. Therefore, the Fe lattices play a dominant role in the angular momentum transfer driven by coherent spin pumping. Neglecting the contribution from Gd in coherent spin pumping due to its weak efficiency, the total damping can be simplified as $\alpha_{tot} = \alpha_{eff0} + \Delta\alpha_1^{SP} \approx \frac{(\alpha_{Fe} + \Delta\alpha_{Fe,1}^{SP})L_{Fe} + \alpha_{Gd}L_{Gd}}{|L_{Fe} - L_{Gd}|}$. Therefore, the enhanced damping $\Delta\alpha_1^{SP}$ driven by coherent spin pumping can be simplified as $\Delta\alpha_1^{SP} = \alpha_{tot} - \alpha_{eff0} \approx \frac{\Delta\alpha_{Fe,1}^{SP}}{|L_{Fe} - L_{Gd}|}$.

Following your comment, we further explained this statement in the revised manuscript [lines 236-242]. A more detailed explanation can be found in the revised SI [lines 176-188].

5. Page 9, line 243, the citation needs brackets.

Response: Thank you. A bracket has been added in the revised manuscript.

Reviewer #3 (Remarks to the Author):

In response to the reports by both reviewers, the authors have revised the manuscript and SI. In the revised manuscript, they now refer to spin-current transmission instead of magnon transmission (it is now clear that they study FMR and not propagating magnons) and clarified previously confusing aspects of the data evaluation regarding the magnon-magnon coupling strength.

With the revisions made by the authors and their detailed reasoning provide in their reply, I believe that I have now been able to understand the physics at play. My previous concerns have thus been adequately addressed and resolved. I particularly appreciate the additions to the SI regarding determination of the coupling strength, which I now find well described and convincing. I now agree with the main conclusions about the observed physical phenomena drawn by the authors. I find that the revised manuscript is a valuable addition to the body of literature in the field. As all-insulating heterostructures are an important platform for spin dynamics, I have the impression

that the topic of the work is suitable for Nature Comm. Most relevant for me, I have no further concerns regarding the interpretation, validity, and discussion of the results.

Response: Thank you so much for your comments on improving our paper and for your recommendation to publish it in Nature Communications.

References

- 1 Kurebayashi, H. *et al.* Controlled enhancement of spin-current emission by three-magnon splitting. *Nat. Mater.* **10**, 660-664 (2011).
- 2 Lee, O. *et al.* Nonlinear Magnon Polaritons. *Phys. Rev. Lett.* **130**, 046703 (2023).
- 3 Arias, R. & Mills, D. Extrinsic contributions to the ferromagnetic resonance response of ultrathin films. *Phys. Rev. B* **60**, 7395 (1999).
- 4 Woltersdorf, G. & Heinrich, B. Two-magnon scattering in a self-assembled nanoscale network of misfit dislocations. *Phys. Rev. B* **69**, 184417 (2004).
- 5 Li, Q. *et al.* Coherent ac spin current transmission across an antiferromagnetic CoO insulator. *Nat. Commun.* **10**, 1-6 (2019).
- 6 Dabrowski, M. *et al.* Coherent Transfer of Spin Angular Momentum by Evanescent Spin Waves within Antiferromagnetic NiO. *Phys. Rev. Lett.* **124**, 217201 (2020).
- 7 Baker, A. A. *et al.* Anisotropic Absorption of Pure Spin Currents. *Phys. Rev. Lett.* **116**, 047201 (2016).
- 8 Li, Y. *et al.* Coherent Spin Pumping in a Strongly Coupled Magnon-Magnon Hybrid System. *Phys. Rev. Lett.* **124**, 117202 (2020).
- 9 Šimánek, E. & Heinrich, B. Gilbert damping in magnetic multilayers. *Phys. Rev. B* **67**, 144418 (2003).
- 10 Heinrich, B. *et al.* Dynamic exchange coupling in magnetic bilayers. *Phys. Rev. Lett.* **90**, 187601 (2003).
- 11 Tserkovnyak, Y., Brataas, A., Bauer, G. E. & Halperin, B. I. Nonlocal magnetization dynamics in ferromagnetic heterostructures. *Rev. Mod. Phys.* **77**, 1375 (2005).
- 12 Takahashi, S. Giant enhancement of spin pumping in the out-of-phase precession mode. *Appl. Phys. Lett.* **104**, 052407 (2014).
- 13 Nat Mater Omelchenko, P., Girt, E. & Heinrich, B. Test of spin pumping into proximity-polarized Pt by in-phase and out-of-phase pumping in Py/Pt/Py. *Phys. Rev. B* **100**, 144418 (2019).
- 14 Klingler, S. *et al.* Spin-Torque Excitation of Perpendicular Standing Spin Waves in Coupled YIG/Co Heterostructures. *Phys. Rev. Lett.* **120**, 127201 (2018).
- 15 Fan, Y. *et al.* Resonant Spin Transmission Mediated by Magnons in a Magnetic Insulator Multilayer Structure. *Adv. Mater.* **33**, e2008555 (2021).
- 16 Cheng, R., Xiao, J., Niu, Q. & Brataas, A. Spin pumping and spin-transfer torques in antiferromagnets. *Phys. Rev. Lett.* **113**, 057601 (2014).
- 17 Kamra, A. & Belzig, W. Spin Pumping and Shot Noise in Ferrimagnets: Bridging Ferro- and Antiferromagnets. *Phys. Rev. Lett.* **119**, 197201 (2017).
- 18 Kamra, A., Troncoso, R. E., Belzig, W. & Brataas, A. Gilbert damping phenomenology for two-

- sublattice magnets. *Phys. Rev. B* **98**, 184402 (2018).
- 19 Stanciu, C. D. *et al.* Ultrafast spin dynamics across compensation points in ferrimagnetic GdFeCo: The role of angular momentum compensation. *Phys. Rev. B* **73**, 220402(R) (2006).
- 20 Geprägs, S. *et al.* Origin of the spin Seebeck effect in compensated ferrimagnets. *Nat. Commun.* **7**, 10452 (2016).

Reviewers' Comments:

Reviewer #2:

Remarks to the Author:

In the revised manuscript, the authors responded to all my concerns. I appreciate that they modified the narration so that the manuscript returns to its physical essence. Now to my understanding, the story is in the heterostructure, there is coupling between FMR modes of the two magnetic layers and interlayer spin pumping. Both of them can be tuned by the temperature-dependent magnetization of GdIG. I think this manuscript may be published in Nature Communications.